# Temporal refinement of Dach1 expression contributes to the development of somatosensory neurons

Tünde Szemes[1,2], Alba Sabaté San José[1], Abdulkader Azouz[3,4], Maren Sitte [5], Gabriela Salinas[5], Younes Achouri [6], Sadia Kricha[1], Laurence Ris[2], Kristy Red-Horse [7,8,9], Eric J Bellefroid [1✉] & Simon Desiderio [1✉]

## Abstract

During somatosensory neurogenesis, neurons are born in an unspecialized transcriptional state. Several transcription factors in these cells follow a broad-to-restricted expression trajectory as development proceeds, giving rise to neuron subtypes with different identities. The relevance of this temporal refinement of transcription factor expression remains unclear as the functions of transcription factors with broad-to-restricted expression patterns have been mostly studied in those neuron subtypes in which they remain active. Here we show that Dach1 encodes a bona fide transcription factor with a broad-to-restricted expression pattern retained and required in tactile somatosensory neurons. In developing nociceptors, Prdm12 contributes to Dach1 silencing. Using genetic approaches to prevent its temporal restriction during mouse somatosensory development, we reveal that Dach1 expression refinement is a prerequisite for the acquisition of an appropriate transcriptional profile in those somatosensory neuron subtypes in which it becomes ultimately silenced. These findings highlight the essential role played by Dach1 during somatosensory neuron development and demonstrate that the temporal pattern of broad-to-restricted expression followed by several transcription factors is physiologically important for the development of somatosensory neurons.

**Keywords** LTMR; Nociceptor; Peripheral Nervous System; Dorsal Root Ganglia
**Subject Categories** Chromatin, Transcription & Genomics; Development; Neuroscience

## Introduction

Primary somatosensory neurons constitute a heterogenous population of peripheral nervous system (PNS) neurons tuned to respond to dedicated stimuli emanating from the external and internal environment. These neurons are found in somatosensory ganglia such as the trigeminal ganglia innervating the head and the dorsal root ganglia (DRG) innervating the body. They support the broad spectrum of somesthesia and are segregated based on their conduction velocity and degree of myelination (Aα-, Aβ-, Aδ- or C subtypes, from heavily myelinated to unmyelinated), their morphology and their transcriptional and physiological features. They can be broadly categorized into three cardinal functional populations; the proprioceptive neurons that innervate tendons and muscles to provide a sense of body position; the low-threshold mechanoreceptors (LTMR) which comprise neurons of C-, Aδ- or Aβ- subtypes involved in touch through the detection of innocuous mechanical stimuli, and the nociceptors comprising a set of largely functionally polymodal neurons of Aδ- or C- subtype involved in nociception, detection of temperature or pruriception (Meltzer et al, 2021).

In the body, these neurons stem from a pool of neural crest cells biased to the somatosensory lineage through the transient expression of the proneural factors Neurog1 and Neurog2 (Meltzer et al, 2021; Vermeiren et al, 2020; Lallemend and Ernfors, 2012). The segregated expression of dedicated neurotrophin receptors later allows a target-derived signal-based crosstalk with the environment to support the survival and maturation of these neurons and drive some aspects of their subtype-specific morphological and transcriptional features (Lallemend and Ernfors, 2012; Moqrich et al, 2004; Marmigère and Ernfors, 2007). Thereby, developing nociceptors initially express the neurotrophin receptor TrkA and later diversify into peptidergic or non-peptidergic nociceptors which respectively maintain TrkA or express Ret (Luo et al, 2007) while proprioceptors express TrkC and A-LTMR

[1]Department of Molecular Biology, ULB Neuroscience Institute (UNI), Université Libre de Bruxelles (ULB), Gosselies, Belgium. [2]Laboratory of Neuroscience, UMONS Research Institute for Health Sciences and Technology, University of Mons, Mons, Belgium. [3]Institute for Medical Immunology (IMI), Université Libre de Bruxelles (ULB), Gosselies, Belgium. [4]ULB Center for Research in Immunology (U-CRI), ULB, Brussels, Belgium. [5]NGS Integrative Genomics, Institute of Pathology at the University Medical Center Göttingen (UMG), 37075 Göttingen, Germany. [6]Transgenesis Platform, de Duve Institute, Université Catholique de Louvain, Brussels, Belgium. [7]Department of Biology, Stanford University, Stanford, CA, USA. [8]Institute for Stem Cell Biology and Regenerative Medicine, Stanford University School of Medicine, Stanford, CA, USA. [9]Howard Hughes Medical Institute, Stanford, CA, USA. ✉E-mail: ebellefr@ulb.be; sdesider@ulb.be

subtypes rely on selective or combinatorial expression of Ret, TrkB or TrkC (Marmigère and Ernfors, 2007; Lallemend and Ernfors, 2012). A general view of neuronal development is that neurons mature through the orderly temporal induction and action of stepwise transcriptional programmes (Faure et al, 2022). Hence, several subtype-restricted transcription factors (TFs) are also widely involved in the development and maturation of these somatosensory neurons at defined periods during their maturation process (Lallemend and Ernfors, 2012; Vermeiren et al, 2020; Hippenmeyer et al, 2005).

While case by case functional studies have proven valuable to understand their selective developmental requirement in somatosensory subtypes, recent findings highlighted that a cohort of these subtype-restricted TF are initially widely co-expressed in newborn somatosensory neurons, following extinction of *Neurog1* and *Neurog2* (Sharma et al, 2020). Newborn somatosensory neurons thus transition from an unspecialized state characterized by wide co-expression of TFs to subsequent stages characterized by the resolution of restricted combinatorial expression of these TFs that transcriptionally define and specify somatosensory subtype identities (Meltzer et al, 2021; Sharma et al, 2020; Rayon et al, 2021). Loss of function studies aimed at identifying the role of genes such as *Pou4f2, Pou4f3, Runx1, Runx3* or *Shox2* which encode for some of these broad-to-restricted TFs have demonstrated that their depletion from the time of their broad expression period results in defects selectively compartmentalized to the somatosensory subtypes in which they are ultimately maintained (Inoue et al, 2002; Levanon et al, 2002; Chen et al, 2006; Qi et al, 2017; Scott et al, 2011; Sharma et al, 2020). These observations raise the question of the biological significance of this broad-to-restricted temporal pattern as TFs showing such a developmental dynamic do not appear to play prominent function at the time of their broad expression period.

Searching to describe new TFs with broad-to-restricted expression in developing DRG, we identified the gene encoding for the helix-turn-helix transcription factor Dach1, which relates to the sno/ski family of corepressors (Mardon et al, 1994). We found that Dach1 is initially ubiquitously expressed in unspecialized postmitotic somatosensory neurons and becomes gradually selectively enriched in TrkB-expressing A-LTMR neurons as development proceeds. In the nociceptive lineage, we demonstrate that this concomitant temporal extinction of Dach1 involves the nociceptor-specific transcriptional regulator Prdm12. Genetic depletion of *Dach1* during its broad or restricted expression period results in A-LTMR restricted defects, thus classifying Dach1 as a bona fide broad-to-restricted TF. Gain-of-function strategies to prevent the temporal refinement of Dach1 result in wide transcriptional defects in neurons that should repress Dach1, reflected at adulthood by altered central innervation and reduced nociception. Altogether, these results establish Dach1 as a new component of the transcriptional pathways leading to the acquisition of an appropriate transcriptional profile in developing somatosensory neurons, for which both the silencing and maintenance of its expression are critical. They further provide clues to the biological significance of the tight temporal regulation of broad-to-restricted TFs as a key event to achieve proper development of somatosensory neurons.

# Results

## Dach1 is expressed during somatosensory neurons development following a broad-to-restricted expression pattern

Recent advances in the understanding of the maturation and diversification of developing and adult somatosensory neurons arose from newly generated single-cell transcriptomic atlases. These databases can be leveraged to highlight uncharacterized genes selectively expressed in discrete somatosensory subtypes (Data ref: Kupari and Ernfors, 2023). Looking for genes encoding transcriptional regulators with such discrete pattern, we identified *Dach1*, a member of the sno/ski family of corepressors, which is putatively enriched at adulthood in myelinated LTMR neurons (Appendix Fig. S1A).

To characterize Dach1 expression during somatosensory neuron development, we first performed double immunostainings for Dach1 and Sox10 to label progenitors on murine embryonic DRG sections. From E10.5 onward, only few Dach1 staining was observed in Sox10[+] somatosensory progenitors (Fig. EV1A). Analysis of single-cell RNA-seq data (Data ref: Sharma et al, 2020; https://kleintools.hms.harvard.edu/tools/springViewer_1_6_dev.html?datasets/Sharma2019/all) from E11.5 *Sox10*-expressing DRG cells led to a similar observation but also did not reveal any clustered expression of Dach1 among *Sox10*-expressing cells (Fig. EV1B). Finally, labelling of E10.5 DRG sections following a 20-min pulse of BrdU revealed that among Sox10[+] cells, the ones co-expressing Dach1 do not incorporate BrdU, suggesting that Dach1 becomes detected in Sox10[+] cells as they transition to a postmitotic state (Fig. EV1C). Afterward, comparison of Dach1 and Islet1, a general marker of postmitotic somatosensory neurons, revealed robust co-expression between E10.5 and E18.5. While most neurons retained Dach1 at early developmental stages, it was widely downregulated at E14.5 (Fig. 1A,B). Analysis of the fluorescence level of Dach1 immunostaining as a readout of its expression in Islet1[+] neurons revealed that Dach1 refinement occurs around E13.5, corresponding to the end of the period of overt neurogenesis in DRG (Meltzer et al, 2021; Vermeiren et al, 2020) (Fig. 1C). This temporal broad-to-restricted expression was also evidenced through analysis of *Dach1*-expressing cells in E11.5, E12.5 and E15.5 DRG in single-cell RNA-seq database (Sharma et al, 2020), which showed that the *Dach1* cluster is enriched in *Ntrk1*/TrkA-expressing nociceptors at E12.5 before switching to mostly *Ntrk2*/TrkB-expressing LTMR at E15.5 (Fig. 1D).

To confirm and further refine the identity of neurons maintaining a strong Dach1 staining at E18.5, double immunostainings were carried out with Prdm12 labelling nociceptors and C-LTMR, TrkB labelling A-LTMR, TrkC labelling A-LTMR and proprioceptors and Parvalbumin (Parvalb) labelling proprioceptors. 86.77% ± 1.94% (Mean ± SEM, n = 3) and 17.48% ± 1.79% of E18.5 Dach1[+] neurons respectively colocalized with TrkB and TrkC. Only a negligible proportion of Dach1[+] neurons colocalized with Prdm12 (5.17% ± 1.127%) or Parvalbumin (1.99% ± 0.18%). These results demonstrated that Dach1 becomes enriched in TrkB-expressing A-LTMR. This was further confirmed by monitoring the co-expression of Dach1 with Ret, MafA and c-Maf, other known markers of A-LTMR subpopulations (Fig. 1E).

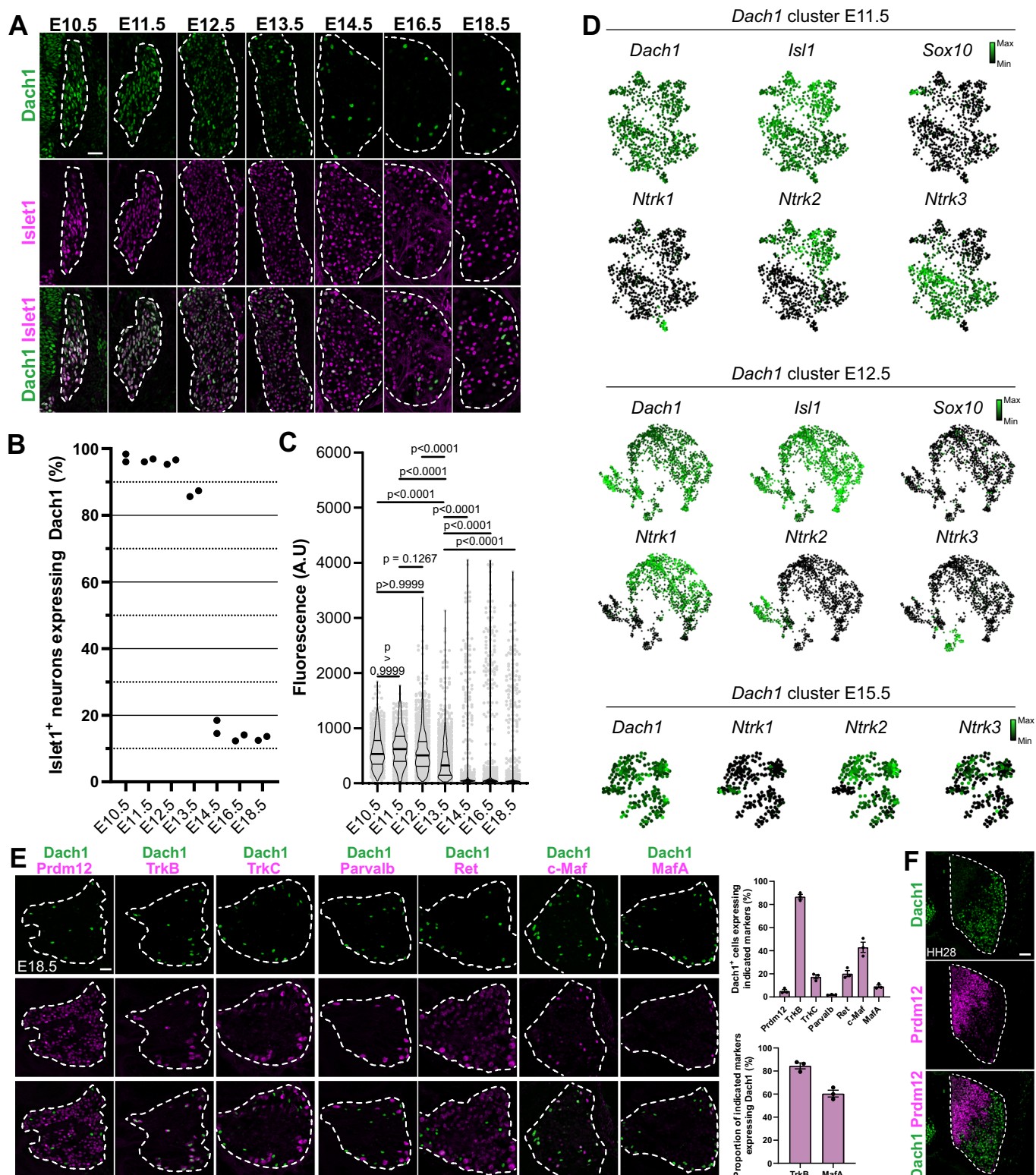

Contrary to what is observed in mice, chicken embryonic DRG show a clear spatial segregation of neuronal subtypes, with nociceptors and mechano/proprioceptors being respectively confined to the dorsomedial and ventrolateral part of the DRG (Rifkin et al, 2000). Co-immunostaining of Prdm12 and Dach1 performed on HH28 chicken DRG sections revealed that Prdm12+ and Dach1+ cells are dorsomedially and ventrolaterally segregated as expected (Fig. 1F). In DRG of adult mice, Dach1 remains preferentially expressed in TrkB-expressing neurons compared to other cardinal subtypes. Single-cell RNA-seq databases indicate that *Dach1* is

**Figure 1. Dach1 follows a broad-to-restricted expression pattern in developing dorsal root ganglia.**

(A) Representative pictures of double immunostainings of Dach1 with the pan-neuronal marker Islet1 performed on thoracic DRG sections of murine embryos at the indicated developmental stages. (B) Scatter dot plots representing the percentage of Islet1+ neurons co-expressing Dach1 in DRG transverse sections from two biological replicates at indicated developmental stages. (C) Quantification of Dach1 fluorescence staining (arbitrary unit, A.U.) among Islet1+ DRG neurons at indicated developmental stages. Each grey point represents the fluorescence level of Dach1 in a single Islet1+ nucleus. The total number of Islet1+ neurons measured at each stage were: $n = 611$ at E10.5, $n = 489$ at E11.5, $n = 1145$ at E12.5, $n = 1186$ at E13.5, $n = 914$ at E14.5, $n = 998$ at E16.5 and $n = 821$ at E18.5. For each developmental stage, all cells whose fluorescence was quantified came from two embryos harvested on a same slide and stained and acquired with same parameters to reduce sample variability. Overlayed violin plots represent data distribution with first and third quartiles indicated by plain lines and median indicated by thick plain line. Kruskal-Wallis test with Dunn's post hoc multiple comparison. (D) Force-directed layout of single-cell RNA-seq data of E11.5, E12.5 and E15.5 DRG neurons found to express *Dach1* overlaid with expression levels of indicated genes. Data were obtained from Sharma et al, 2020. (E) Left, double immunostainings with Dach1 and indicated markers performed on E18.5 thoracic DRG transverse sections. Right, quantifications of the colocalization of Dach1 and indicated somatosensory subtype markers. Data are represented as mean ± SEM, three embryos analysed per condition. (F) Double immunostaining of Dach1 and Prdm12 on transverse DRG section of embryonic chicken at Hamburger & Hamilton (HH) stage 28. DRG in pictures are delineated by white dashed lines. Scale bars, 50 µm. Source data are available online for this figure.

notably enriched in TrkB+ Aδ-LTMR. This observation was further confirmed in situ by analysing the colocalization of Dach1 with the Aδ-LTMR marker *Colq* (Sharma et al, 2020) (Appendix Fig. S1B,C). These findings demonstrate that Dach1 follows a broad-to-restricted expression pattern in developing DRG to become progressively compartmentalized to Aβ- and Aδ-LTMR.

## The nociceptor determinant Prdm12 contributes to the temporal refinement of Dach1

The temporal restriction of Dach1 to A-LTMR neurons implies its progressive repression by other somatosensory subtypes, including by nociceptors, which represent the vast majority of developing somatosensory neurons (Vermeiren et al, 2020). This temporal extinction was evidenced through co-immunostaining of Dach1 with Prdm12 and TrkA, two general markers of developing nociceptors, before (E12.5) and after (E16.5) its temporal refinement (Fig. 2A). Previous reports suggest that neurotrophin-derived signalling pathways are involved in some aspects of the transcriptional refinement controlling subtype identity. Among them, the NGF-TrkA signalling pathway that plays a predominant role in the embryonic survival and development of the nociceptive lineage (Sharma et al, 2020; Luo et al, 2007).

As most neurons downregulating Dach1 are inferred to be developing nociceptors, we thus analysed Dach1 expression in DRG of $NGF^{-/-}; Bax^{-/-}$ E18.5 embryos compared to control ones. While Ret expression was abolished in nociceptors of $NGF^{-/-}; Bax^{-/-}$ mutants as previously reported (Luo et al, 2007), Dach1 expression was not broadened, indicating that Dach1 temporal restriction does not depend on the NGF-TrkA signalling (Fig. 2B). In DRG of $NGF^{-/-}; Bax^{-/-}$ E18.5 embryos, we also analysed the expression of the transcriptional regulator Prdm12, a critical determinant of the development of nociceptors whose invalidation results in agenesis of the entire nociceptive lineage (Desiderio et al, 2019; Bartesaghi et al, 2019; Landy et al, 2021; Kokotović et al, 2021). Figure 2C shows that Prdm12 is still expressed in DRG of $NGF^{-/-}; Bax^{-/-}$ E18.5 embryos suggesting that its expression, like that of Dach1, is independent of NGF. Prdm12 is mainly known to act as a putative transcriptional repressor (Thélie et al, 2015). In developing DRG, its loss results in the ectopic expression of visceral neuron determinants in a portion of putative nociceptive precursors (Vermeiren et al, 2023). This prompted us to hypothesize that it may be involved in Dach1 refinement. However, given the combination of rapid cell loss and ectopic Phox2a/b expression in newborn nociceptors constitutively lacking *Prdm12*, unambiguous analysis of Dach1 refinement in DRG

of *Prdm12* KO embryos is challenging. To overcome this difficulty, we generated $Scn10a^{Cre/+}; Prdm12^{flox/flox}$ (*Prdm12* cKO) mice which result in late post-mitotic embryonic depletion of *Prdm12* in developing nociceptors without cell loss or ectopic expression of *Phox2a/b* (Vermeiren et al, 2023). Analysis of DRG of E18.5 *Prdm12* cKO embryos revealed broadened expression of Dach1 (Fig. 2D,E). Analysis of the pain response of these *Prdm12* cKO mice to the formalin inflammatory pain assay further revealed an impaired behavioural response. These observations suggest that the defective embryonic extinction of *Dach1* could contribute to the reduced pain response observed in *Prdm12* cKO mice (Fig. 2F,G).

To further investigate the mechanism involving *Dach1* repression by Prdm12, we sought to perform chromatin immunoprecipitation (ChIP) experiments to determine whether *Dach1* may represent a direct transcriptional target of Prdm12. As no ChIP-grade Prdm12 antibodies have been described so far, we generated a *Prdm12* knock-in (KI) mouse line in which *Flag* and *V5* coding sequences were inserted in frame downstream of the endogenous *Prdm12* coding sequence, resulting in the production of a Prdm12 protein fused to three Flag and one V5 tags in its C-terminal region (*Prdm12KI*, Fig. 2H). *Prdm12KI/KI* mice were viable, fertile and raised to adulthood without obvious abnormalities contrary to the birth lethal phenotype of $Prdm12^{-/-}$ mice (Desiderio et al, 2019), indicating that addition of the tags does not preclude Prdm12 function. In the developing nervous system, *Prdm12* is notably found in spinal cord p1 progenitors (Thélie et al, 2015). Accordingly, immunostainings targeting Prdm12 and V5 epitopes performed on E12.5 *Prdm12KI/+* spinal cord and DRG transverse sections indicated that V5 faithfully recapitulates Prdm12 expression in both structures (Fig. 2I,J).

ChIP was subsequently applied to *Prdm12KI/+* and *Prdm12+/+* E12.5 isolated spinal cords with attached DRG or E13.5 isolated DRG using a V5 antibody and next-generation sequencing was performed on harvested DNA to identify Prdm12 bound regions. A previous study reported that in *Xenopus laevis* animal cap explants treated with retinoic acid and overexpressing *Noggin* and *Prdm12*, Prdm12 binds downstream of the homeobox gene *Dbx1*, one of its repressed targets during spinal cord neurogenesis (Thélie et al, 2015). Mapping this genomic bound region in the resulting dataset to the murine genome, we consistently observed a specific enrichment of Prdm12 binding in a genomic region downstream of the murine *Dbx1* locus that is homologous to the one previously reported in *Xenopus laevis* (Fig. 2K). This result was further validated by ChIP-qPCR, confirming that our ChIP-seq successfully identified Prdm12 bound genomic regions (Fig. 2L). Analysis of the ChIP-seq dataset around the *Dach1* locus

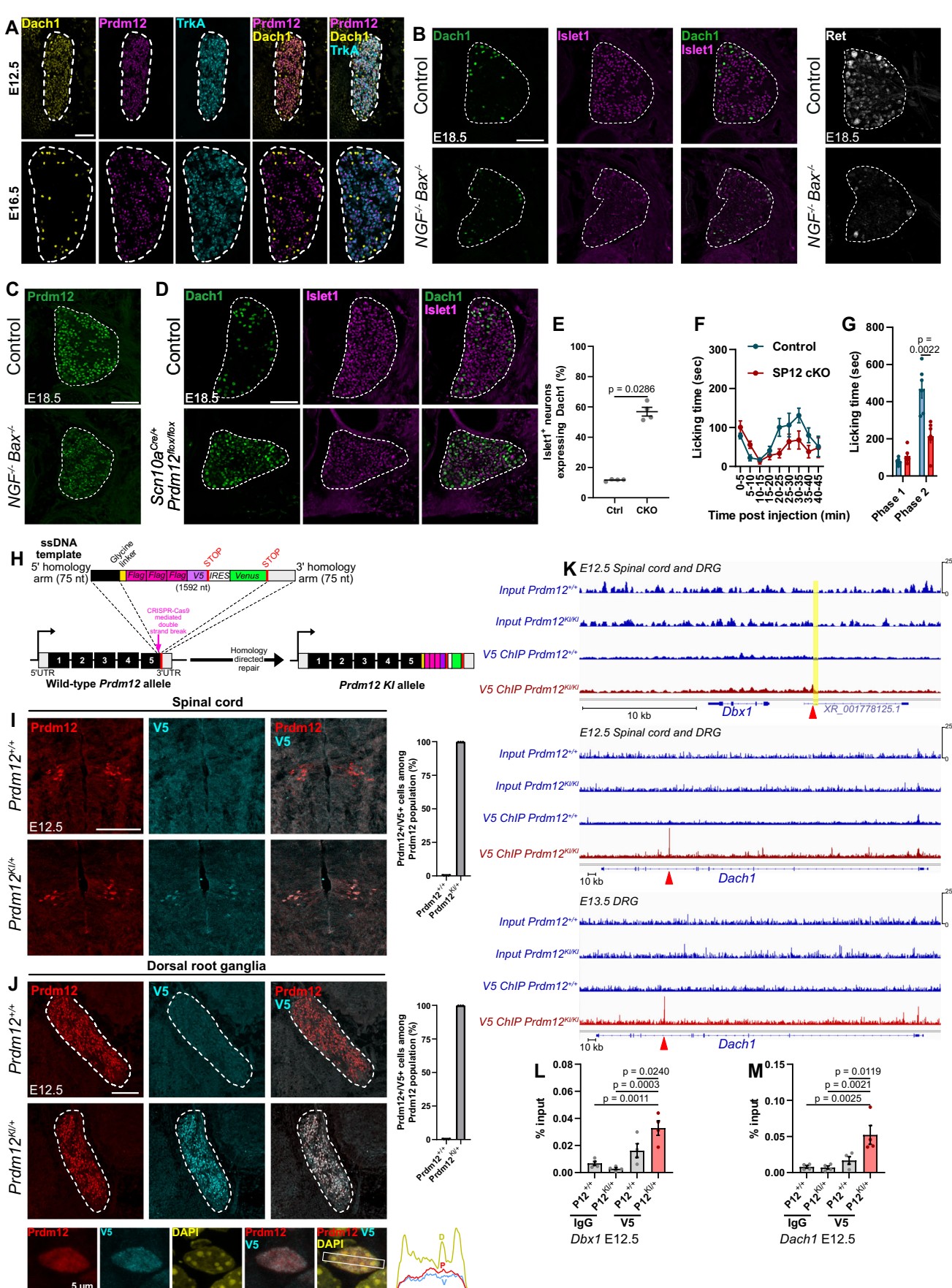

**Figure 2. Prdm12 contributes to the temporal refinement of Dach1 independently of NGF signalling.**

(A) Triple immunostaining of Dach1, Prdm12 and TrkA performed on E12.5 and E16.5 DRG transverse sections of control embryos. (B) Left, double immunostaining of Dach1 and Islet1 performed on E18.5 DRG transverse sections of control and $NGF^{-/-};Bax^{-/-}$ embryos. Note the maintenance of Dach1 restricted expression pattern in absence of *NGF*. Right, Immunostaining targeting Ret performed as a positive control showing the proper inactivation of the NGF signalling pathway in $NGF^{-/-};Bax^{-/-}$ E18.5 DRG. (C) Immunostaining targeting Prdm12 performed on E18.5 DRG transverse sections of control and $NGF^{-/-};Bax^{-/-}$ embryos. Note the maintained expression of Prdm12 in absence of NGF. (D) Double immunostaining of Dach1 and Islet1 performed on E18.5 DRG transverse sections of control and $Scn10a^{Cre/+}; Prdm12^{flox/flox}$ embryos. Note the broadening of Dach1$^+$ cells in absence of Prdm12. (E) Quantification analysis of the mean percentage of Islet1$^+$ neurons expressing Dach1 in DRG of E18.5 control and $Scn10a^{Cre/+}; Prdm12^{flox/flox}$ (CKO) embryos. Mann–Whitney tests. Data are presented as the mean ± SEM of quantifications from four embryos per condition. (F) Time course nocifensive response (licking time) of control ($n = 6$) and SP12 cKO ($n = 6$) littermates until 45 min following formalin injection. Data are presented as the mean ± SEM. Two-way ANOVA with Fisher's LSD post hoc test. (G) Scatter dot plot showing the response of individuals of indicated genotypes during the first (0 to 5 min following injection) and the second (15 to 45 min following injection) phase of nocifensive behaviour of the formalin assay. Mann–Whitney tests. Data are presented as the mean ± SEM of quantifications from six embryos per condition. (H) Schematic representation of the CRISPR-Cas9-based strategy used to generate the $Prdm12^{KI}$ allele. (I) Left, double immunostaining of Prdm12 and V5 performed on E12.5 spinal cord transverse sections of $Prdm12^{+/+}$ and $Prdm12^{KI/+}$ embryos. Right, quantification analysis of the mean percentage of V5$^+$ nuclei expressing Prdm12 in E12.5 spinal cords of $Prdm12^{+/+}$ and $Prdm12^{KI/+}$ embryos. Data are presented as the mean ± SEM of quantifications from three embryos per condition. (J) Left, double immunostaining of Prdm12 or TrkB with V5 performed on E12.5 DRG transverse sections of $Prdm12^{+/+}$ and $Prdm12^{KI/+}$ embryos or $Prdm12^{KI/+}$ embryos alone. Top right, quantification analysis of the mean percentage of V5$^+$ nuclei expressing Prdm12 in E12.5 DRG of $Prdm12^{+/+}$ and $Prdm12^{KI/+}$ embryos. Data are presented as the mean ± SEM of quantifications from three embryos per condition. Bottom right, quantification analysis of the mean percentage of V5$^+$ nuclei found in TrkB$^+$ neurons in E12.5 DRG of $Prdm12^{KI/+}$ embryos. Bottom, colocalization analysis of Prdm12 and V5 in a E12.5 DRG cell counterstained with DAPI (scale bar, 5 μm). (K) Mouse genomic region surrounding *Dbx1* (top) or *Dach1* (middle) showing sequencing data for input DNA recovered from $Prdm12^{+/+}$ and $Prdm12^{KI/+}$ and V5 ChIP-enriched DNA recovered from $Prdm12^{+/+}$ and $Prdm12^{KI/+}$ E12.5 spinal cord with attached DRG. Bottom trace view shows that the peak observed at E12.5 is still found following V5 targeted ChIP-seq performed on isolated E13.5 DRG. Peaks of interest are highlighted by red arrowheads. The homologous Prdm12 bound region near *Dbx1* previously characterized in *Xenopus laevis* is highlighted by a yellow band. (L) ChIP-qPCR validation of the peak identified in the vicinity of the *Dbx1* locus performed on dissected spinal cord and DRG of E12.5 embryos of indicated genotypes. Graphical data in this panel are presented as mean ± SEM of four biological replicates. One way ANOVA test with Dunnett's post hoc multiple comparison. (M) ChIP-qPCR validation of the peak identified in the fourth intron of the *Dach1* gene performed on dissected spinal cord and DRG of E12.5 embryos of indicated genotypes. Graphical data in this panel are presented as mean ± SEM of four biological replicates. One way ANOVA test with Dunnett's post hoc multiple comparison. DRG in pictures are delineated by white dashed lines. Scale bars, 100 μm. Source data are available online for this figure.

highlighted an intronic Prdm12 binding site in the *Dach1* gene both in the E12.5 spinal cord with associated DRG dataset and in the E13.5 isolated DRG dataset, indicating that the binding indeed occurs in somatosensory neurons (Fig. 2K). Binding to this putative regulatory region was further validated by ChIP-qPCR (Fig. 2M). Together, these results indicate that Prdm12 contributes to the temporal restriction of *Dach1* in the nociceptive lineage, independently of the NGF pathway.

## Loss of Dach1 results in tactile defects without affecting somatosensory neurogenesis

To evaluate the involvement of *Dach1* in somatosensory neuron development, we relied on two strategies aiming at depleting *Dach1* during (I) its early broad expression period or (II) its A-LTMR restricted expression period. To achieve this purpose, mice carrying a *Dach1* floxed exon 2 ($Dach1^{flox/flox}$) were crossed with $Wnt1^{Cre}$ or $Advillin^{Cre}$ mice to respectively deplete *Dach1* from the neural crest or from postmitotic somatosensory neurons (Chang et al, 2017; Hasegawa et al, 2007; Rowitch et al, 1998). While $Advillin^{Cre/+}$; $Dach1^{flox/flox}$ conditional knockout (*AD1* cKO) mice are viable and fertile, $Wnt1^{Cre/+}; Dach1^{flox/flox}$ (*WD1* cKO) mice die soon after birth, suggesting that *Dach1*-associated neural crest defects contribute to the postnatal lethality previously reported in *Dach1* constitutive knockouts (Davis et al, 2001). Immunostaining of Dach1 confirmed its extinction in both lines by E18.5 (Appendix Fig. S2A). Furthermore, Dach1 was already abolished by E12.5 in DRG of *WD1* cKO embryos, indicating that its depletion occurs as expected during its pan-sensory period (Appendix Fig. S2B). In contrast, Dach1 depletion in *AD1* cKO embryos became effective around E14.5, when Dach1 is predominantly restricted to TrkB$^+$ A-LTMR (Appendix Fig. S2C–E). In *WD1* cKO and *AD1* cKO E18.5 embryos, the counting of Islet1$^+$ somatosensory neurons in thoracic DRG revealed no difference between mutants and controls. We also examined bona fide general markers of somatosensory subtypes

and found again no difference between the cohorts. Overall, these results show that depletion of Dach1 does not impact neurogenesis and does not alter the proportions of the main (nociceptors, LTMR, proprioceptors) neuron subtypes (Fig. 3A).

In adult *AD1* cKO mice, we also analysed histologically the innervation of the mechanosensory end organs of the skin by A-LTMR (Handler and Ginty, 2021). This comprised analysis of LTMR endings innervating hair follicles, Meissner and Pacinian corpuscles and Merkel cells. No terminal morphology or innervation defects were observed in *AD1* cKO compared to control samples (Appendix Fig. S3A–E). Thus, loss of *Dach1* from its restricted period of expression does not result in peripheral innervation defects.

We next wanted to determine whether the loss of *Dach1* affect the response of mice to tactile stimuli. We therefore performed Von Frey and sticky tape tests on *AD1* cKO and control mice. Despite their unaltered A-LTMR peripheral innervation, *AD1* cKO mice displayed reduced tactile sensitivity. We also performed Rotarod and Formalin tests, respectively to challenge kinesthesis and pain. No difference was observed between *AD1* cKO and controls indicating that loss of *Dach1* does not alter proprioception and nociception (Fig. 3B). Altogether, these results demonstrate that despite somatosensory neurons are appropriately generated in the absence of *Dach1*, its expression in A-LTMR is required for appropriate tactile sensitivity.

## The developmental loss of Dach1 primarily affects TrkB$^+$ A-LTMR

While the loss of Dach1 during its refined temporal period using *AD1* cKO mice revealed its selective requirement for touch, such a behavioural analysis is precluded when Dach1 is depleted from its broad expression period due to the postnatal lethality of *WD1* cKO mice. To determine whether early loss of Dach1 in the *WD1* cKO DRG results in transcriptional misregulation selectively in LTMR or,

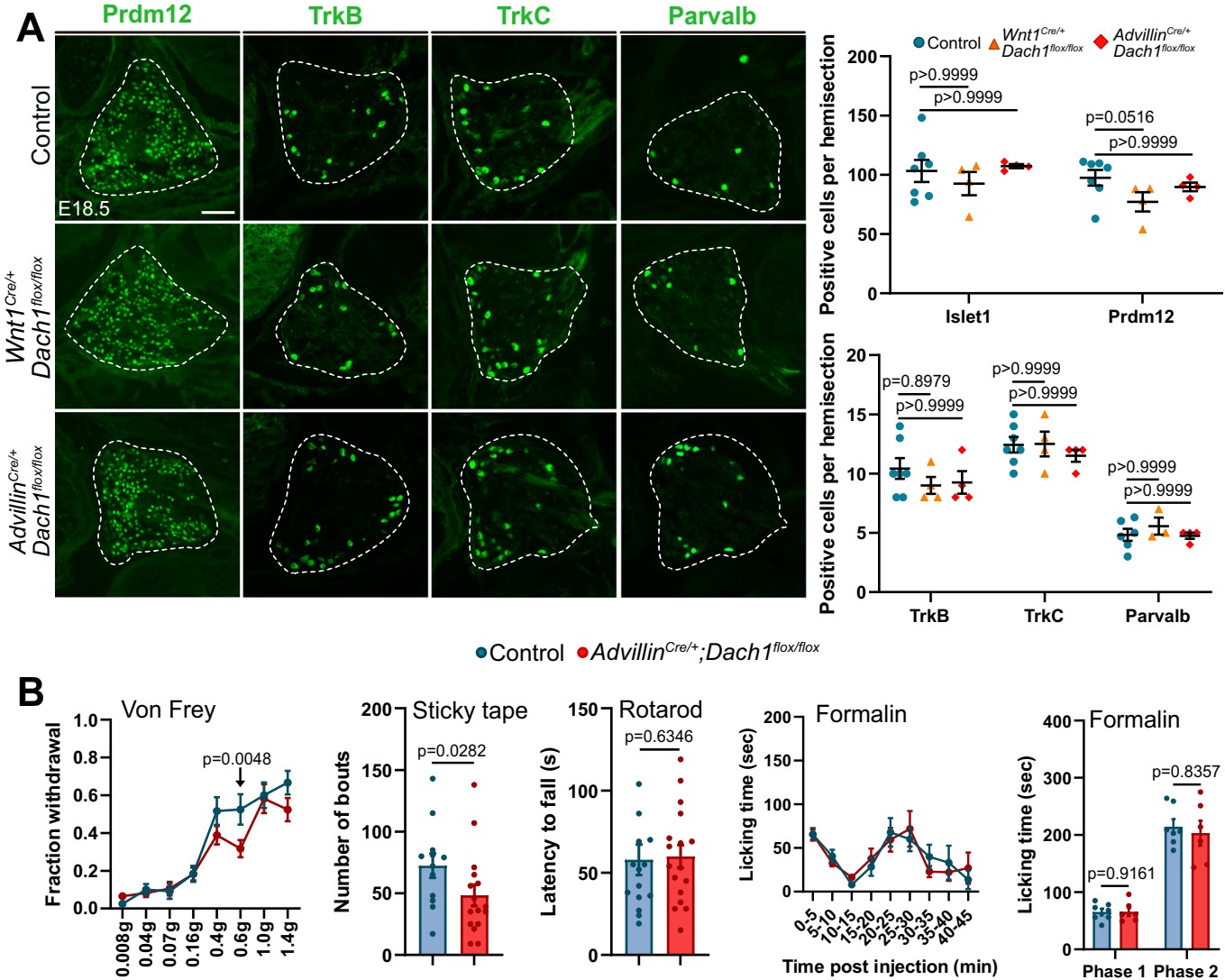

**Figure 3. The loss of Dach1 in somatosensory neurons selectively impairs the sense of touch without affecting neurogenesis.**

(A) Left, Representative pictures of immunostaining targeting indicated somatosensory subtype markers performed on thoracic transverse DRG sections of control, *Wnt1Cre/+; Dach1flox/flox* (*WD1* cKO) or *AdvillinCre/+; Dach1flox/flox* (*AD1* cKO) E18.5 embryos. Right, Quantification analysis represented as scatter dot plot comparing the mean number of neurons immunostained for indicated marker on transverse hemisections of indicated genotypes. Each dot in this scatter plot indicates the mean value obtained for a single embryo (Control, n = 7; WD1 cKO, n = 4; AD1 cKO, n = 4). Graphical data are presented as mean ± SEM. Kruskal-Wallis test with Dunn's post hoc multiple comparison. (B) Behaviour analyses comparing response of control and *AdvillinCre/+; Dach1flox/flox* genotypes to tactile (Von Frey, sticky tape), locomotor (rotarod) and pain (formalin) tests. Von Frey, the response rate (fraction withdrawal) of control (n = 12) and *AdvillinCre/+; Dach1flox/flox* (n = 17) littermates to indicated forces applied to the plantar hindpaw using calibrated Von Frey filaments is indicated. Two-way ANOVA with Fisher's LSD post hoc test. Sticky tape, Quantification of the number of responses to the tactile stimulation of a sticky tape applied on the back skin of control (n = 11) and *AdvillinCre/+; Dach1flox/flox* (n = 16) littermates. Mann–Whitney test. Rotarod, Quantification of the time spent by control (n = 14) and *AdvillinCre/+; Dach1flox/flox* (n = 17) littermates on an accelerating rotarod before falling. Mann–Whitney test. Formalin (left), Time course nocifensive response (licking time) of control (n = 7) and *AdvillinCre/+; Dach1flox/flox* (n = 6) littermates until 45 min following formalin injection. Two-way ANOVA with Fisher's LSD post hoc test. Formalin (right), scatter dot plot indicating the response of individuals of indicated genotypes during the first (0 to 5 min following injection) and the second phase (15 to 45 min following injection). Mann–Whitney test. Each dot in the scatter plots reports the measured value for a single individual. Data in all graphs are represented as mean ± SEM. DRG in pictures are delineated by white dashed lines. Scale bars, 100 μm. Source data are available online for this figure.

conversely, across different neuronal subtypes, a bulk RNA-sequencing of DRG harvested from E18.5 *WD1* cKO and littermate controls was performed (Dataset EV1). Misregulated genes showed only modest fold changes, indicating that early depletion of *Dach1* has only discrete consequences on somatosensory neurons. Using a cut-off of log₂(Fold Change) > |±0.15| with a *p*-value below 0.005, 51 genes were found to be differentially expressed between *WD1* cKO and

controls, of which 24 and 27 were respectively reduced and increased in the *WD1* cKO embryos (Fig. 4A). Despite these gene changes were evidenced at E18.5, analysis of the putative endogenous expression of downregulated genes at adulthood (Data ref: Kupari and Ernfors, 2023) highlighted their prospective preferential enrichment in TrkB⁺ A-LTMR neuron classes (Aβ-RA-LTMR and Aδ-LTMR). Moreover, comparison of the endogenous expression of these genes among

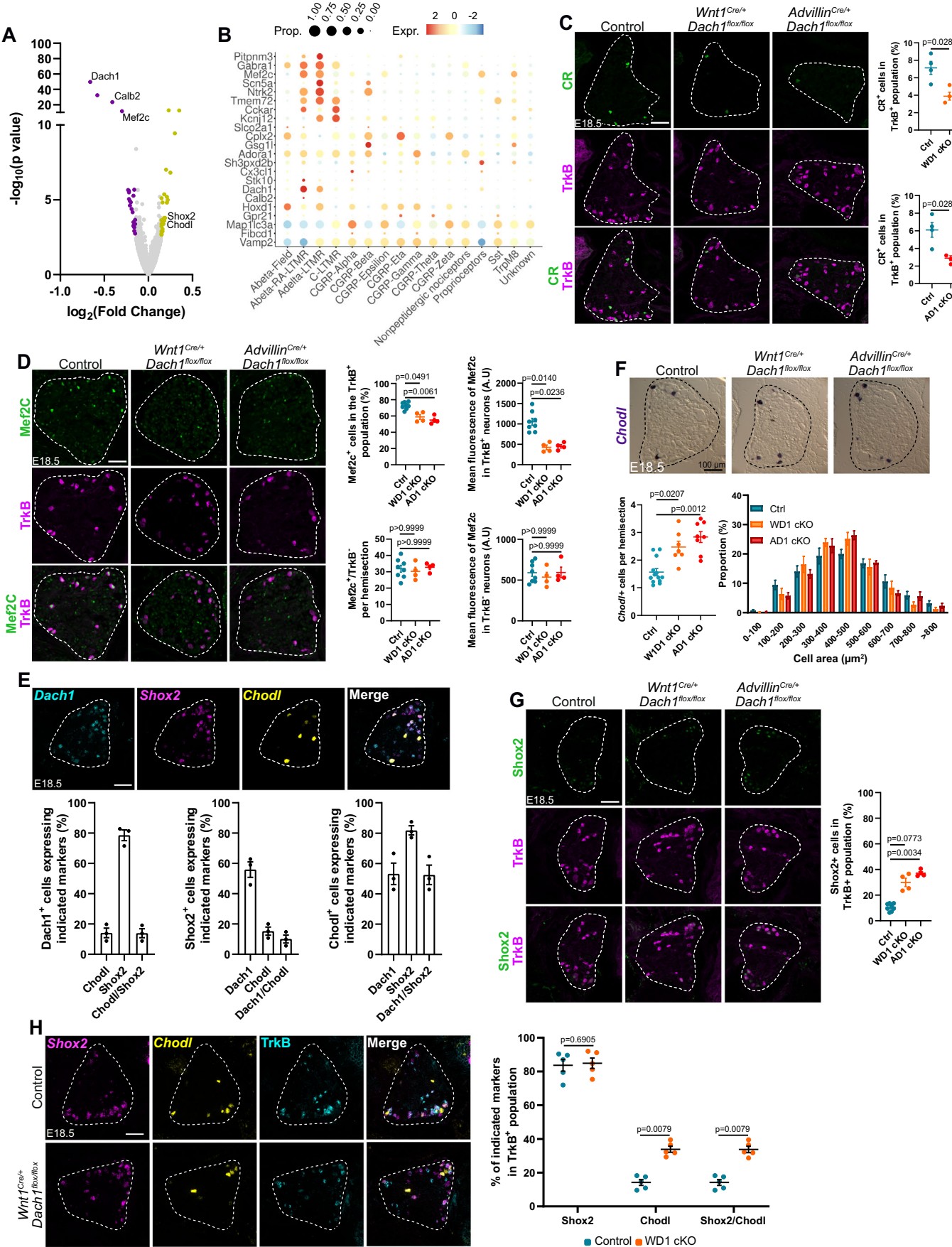

**Figure 4. The loss of Dach1 selectively affects LTMR neurons.**

(A) Volcano plot showing the $-\log_{10}(p\text{-value})$ as a function of $\log_2$(Fold Change) of deregulated genes between DRG of E18.5 control ($n = 5$) and *WD1* cKO ($n = 5$) embryos. Genes were identified by bulk RNA-seq and selected based on a $p$-value $< 0.005$ and a $\log_2$(Fold Change) $> |\pm 0.15|$. genes downregulated and upregulated above this threshold are respectively labelled in purple and yellow. (B) Bubble plot reporting the putative fraction of adult somatosensory subtypes that endogenously express the genes found misregulated in *WD1* cKO E18.5 DRG (generated using https://ernforsgroup.shinyapps.io/MouseDRGNeurons/; Data ref: Kupari and Ernfors, 2023). (C) Left, representative pictures of double immunohistochemistry targeting Calretinin (CR) and TrkB, performed on thoracic transverse DRG sections of control, *WD1* cKO and *AD1* cKO E18.5 embryos. Right, Quantification analysis of the mean percentage of CR$^+$ neurons in the TrkB$^+$ population in DRG of E18.5 embryos of indicated genotypes. Graphical data are presented as a scatter plot reporting mean $\pm$ SEM where each dot represents the mean value of one biological replicate (Control, $n = 4$; *WD1* cKO, $n = 4$; *AD1* cKO, $n = 4$). Mann–Whitney tests. (D) Left, representative pictures of double immunohistochemistry targeting Mef2C and TrkB, performed on thoracic transverse DRG sections of control, *WD1* cKO and *AD1* cKO E18.5 embryos. Right, Quantification analyses of the mean percentage of Mef2C$^+$ neurons inside or outside the TrkB$^+$ compartment and quantification analyses of the mean fluorescence level (arbitrary unit, A.U.) of Mef2C signal inside or outside the TrkB$^+$ compartment, in DRG of E18.5 embryos of indicated genotypes. Graphical data are presented as a scatter plot reporting mean $\pm$ SEM where each dot represents the mean value of one biological replicate (Control, $n = 8$, *WD1* cKO, $n = 4$; *AD1* cKO, $n = 4$). Kruskal-Wallis tests with Dunn's post hoc multiple comparison. (E) Upper panel, representative pictures of thoracic transverse DRG sections of control E18.5 embryos stained following fluorescent in situ hybridization using probe targeting *Dach1*, *Shox2* and *Chondrolectin* (*Chodl*) transcripts. Lower panels, quantification analysis reporting the percentage of *Dach1*$^+$, *Shox2*$^+$ or *Chodl*$^+$ cells co-expressing indicated markers. Graphical data are presented as a scatter plot reporting mean $\pm$ SEM where each dot represents the mean value of one biological replicate ($n = 3$). (F) Upper panel, representative pictures of thoracic transverse DRG sections of control, *WD1* cKO and *AD1* cKO E18.5 embryos stained following in situ hybridization using a probe targeting *Chondrolectin* (*Chodl*) transcripts. Lower left, Quantification analysis comparing the mean number of neurons stained for *Chodl* on transverse hemisections of E18.5 embryos of indicated genotypes. Graphical data are presented as a scatter plot reporting mean $\pm$ SEM where each dot represents the mean value of one biological replicate (Control, $n = 13$; *WD1* cKO, $n = 7$; *AD1* cKO, $n = 8$). Kruskal-Wallis test with Dunn's post hoc multiple comparison. Lower right, histograms representing the percentage of *Chodl*-expressing cells of indicated range of soma area ($\mu m^2$) observed in DRG sections of E18.5 embryos of indicated genotypes. Graphical data are presented as histograms reporting mean $\pm$ SEM (Control, $n = 13$; *WD1* cKO, $n = 7$; *AD1* cKO, $n = 8$). (G) Left, representative pictures of double immunohistochemistry targeting Shox2 and TrkB, performed on thoracic transverse DRG sections of control, *WD1* cKO and *AD1* cKO E18.5 embryos. Right, Quantification analysis of the mean percentage of Shox2$^+$ neurons in the TrkB$^+$ population in DRG of E18.5 embryos of indicated genotypes. Graphical data are presented as a scatter plot reporting mean $\pm$ SEM where each dot represents the mean value of one biological replicate (Control, $n = 8$, *WD1* cKO, $n = 4$; *AD1* cKO, $n = 4$). Kruskal-Wallis tests with Dunn's post hoc multiple comparison. (H) Left, representative pictures of immunohistochemistry targeting TrkB coupled with fluorescent in situ hybridization targeting *Shox2* and *Chodl*, performed on thoracic transverse DRG sections of control and *WD1* cKO E18.5 embryos. Right, quantification analysis of the mean percentage of TrkB$^+$ neurons co-expressing *Shox2*, *Chodl* or both. Graphical data are presented as a scatter plot reporting mean $\pm$ SEM where each dot represents the mean value of one biological replicate (Control, $n = 5$, *WD1* cKO, $n = 5$). Mann–Whitney tests. DRG in pictures are delineated by dashed lines. Scale bars, 100 $\mu m$. Source data are available online for this figure.

*Sox10*-expressing cells at E11.5 and among *Dach1*-expressing neurons at P0 (Sharma et al, 2020; https://kleintools.hms.harvard.edu/tools/springViewer_1_6_dev.html?datasets/Sharma2019/all) indicated that the identified downregulated genes are endogenously enriched in Dach1-expressing LTMR neurons compared to somatosensory progenitors, suggesting that these genes relate to the post-mitotic stage (Fig. EV2). This enrichment, correlated with the subsequent compartmentalized expression of Dach1, suggests its depletion during its broad expression period affects discrete subtypes of somatosensory neurons (Fig. 4B).

Among the most downregulated genes found in our RNA-seq analysis were *Calb2*, a gene encoding for the neuronal modulator Calretinin (CR) found in Aβ-LTMR neurons innervating Pacinian corpuscles (Duc et al, 1994) and *Mef2C*, a gene encoding for a transcription factor deeply involved in the developing central nervous system whose function remains unknown in the developing PNS (Li et al, 2008; Mayer et al, 2018). The downregulation of proteins encoded by those genes was validated through co-immunostainings with TrkB at E18.5, comparing controls to both *WD1* cKO and *AD1* cKO conditions. In controls, CR was selectively found in a small population of TrkB-expressing neurons. Quantification of the percentage of CR$^+$/TrkB$^+$ neurons within the TrkB$^+$ population confirmed its significant reduction in both cKO lines (Fig. 4C). Mef2C similarly showed strong expression within the TrkB$^+$ population but was also expressed in TrkB-negative neurons. Separate analysis of Mef2C expression inside and outside the TrkB$^+$ compartment found no difference in the number of Mef2C$^+$/TrkB$^-$ cells and a modest but significant decrease of Mef2C$^+$/TrkB$^+$ neurons. Beside this reduction, the fluorescence signal of Mef2C in the remaining Mef2C$^+$/TrkB$^+$ neurons appeared consistently reduced in both cKO samples compared to controls. Quantification of the fluorescence level of this remaining Mef2C$^+$

staining as a readout of Mef2C protein abundance in the nucleus confirmed its selective decrease in TrkB$^+$ neurons (Fig. 4D).

We then analysed the expression pattern of *Chodl* and *Shox2*, which were upregulated following *Dach1* depletion. In control DRG, *Shox2* is endogenously found in $78.6 \pm 3.5\%$ (mean $\pm$ SEM) of *Dach1*$^+$ cells while *Chodl* is found in $14.12 \pm 3.2\%$ (mean $\pm$ SEM) of *Dach1*$^+$ cells. Their expression is however not restricted to *Dach1* as *Dach1*$^+$/*Shox2*$^+$ and *Dach1*$^+$/*Chodl*$^+$ cells constitute $55.9 \pm 5.1\%$ (mean $\pm$ SEM) and $53.2 \pm 7.1\%$ (mean $\pm$ SEM) of the respective *Shox2* and *Chodl* populations (Fig. 4E). In situ hybridization of *Chodl* revealed its endogenous restricted expression in a subset of large diameter somatosensory neurons, consistent with its previously reported expression in proprioceptors and in a subset of LTMR (Wu et al, 2021; Sakurai et al, 2013). Analysis of *Chodl* revealed an increase in the number of *Chodl*-expressing cells in DRG of *WD1* cKO and *AD1* cKO E18.5 compared to controls (Fig. 4F). *Chodl*-expressing cells increasingly happened to have a diameter between 300 and 500 $\mu m^2$ in cKO compared to controls, suggesting that ectopic expression of *Chodl* selectively occurs in A-type large diameter neurons (Luo et al, 2007). Analysis of Shox2 by immunostaining revealed a reduced number of detectable positive nuclei compared to RNA expression analysis, a phenomenon previously observed that may reflect the complex post-transcriptional regulation of *Shox2* during sensory neuron development (Peng et al, 2018; Abdo et al, 2011). However, an increased proportion of Shox2$^+$/TrkB$^+$ neurons were found in DRG of E18.5 *WD1* cKO and *AD1* cKO embryos compared to controls (Fig. 4G). Finally, fluorescent in situ hybridization analysis performed on E18.5 DRG sections of control and *WD1* cKO embryos indicated that while the number of *Shox2*-expressing cells is not increased, increased expression of *Chodl* is found in TrkB$^+$/*Shox2*$^+$ neurons (Fig. 4H). In conclusion, the loss of Dach1 at E14.5 using *AD1* cKO mice results in a phenotype similar to the one revealed in *WD1* cKO mice. These

findings further suggest that the loss of *Dach1* during its broad or restricted period primarily results in transcriptional defects associated with TrkB$^+$ A-LTMR neurons. This together with its temporal expression dynamics establishes Dach1 as a bona fide broad-to-restricted transcription factor (Sharma et al, 2020).

## The temporal refinement of Dach1 is a prerequisite for the different somatosensory subtypes to acquire an appropriate transcriptional profile

Depletion of *Dach1* during its broad expression period results in transcriptional defects clustered to the subpopulations in which it remains ultimately maintained. This observation prompted us to investigate the biological relevance of this temporal refinement. In other words, if Dach1 is anyway selectively involved in neurons in which it is maintained, is its repression essential in neurons in which it becomes turned off? To answer that question, we defined a strategy to sustain the pan-sensory expression of Dach1 by taking advantage of a previously described *Rosa26$^{lox-stop-lox-Dach1-IRES-EGFP}$* mouse line which allows spatially and temporally controlled overexpression (OE) of *Dach1* following Cre-mediated excision (Raftrey et al, 2021). We crossed this mouse line with *Advillin$^{Cre}$* mice to generate *Advillin$^{Cre/+}$; Rosa26$^{lox-stop-lox-Dach1-IRES-EGFP/+}$* (*AD1* OE) embryos (Appendix Fig. S4A). Pups of the desired genotype die around birth but otherwise develop until E18.5. At E13.5, a salt and pepper Dach1 expression pattern comparable to the control is observed (Appendix Fig. S4B,C). Nevertheless, analysis of E18.5 *AD1* OE DRG revealed that Dach1 becomes effectively expressed in all somatosensory neurons later in development (Appendix Fig. S4D). Thus, in *AD1* OE embryos, Dach1 is first appropriately refined before being broadly re-expressed at later stage (Appendix Fig. S4E).

To start to characterize the consequences of preventing the temporal refinement of Dach1 in *AD1* OE embryos, we first quantified the number of Islet1$^+$ neurons in DRG of *AD1* OE and control E18.5 embryos. We also analysed the proportion of neurons expressing Prdm12, TrkB and TrkC (Fig. 5A,B). These analyses indicated that the different main subtypes of neurons are produced in appropriate numbers despite the pan-sensory maintenance of Dach1. Nevertheless, the possibility remains that Dach1 temporal refinement stays as a critical event for the subsequent steps of somatosensory neurons development. To further investigate this hypothesis, we collected DRG from E18.5 *AD1* OE and control embryos to perform a bulk RNA-seq analysis (Dataset EV2). Considering the selected differentially expressed genes following a cut-off of log$_2$(Fold Change) > |±0.25| with an adjusted *p*-value below 0.005, 614 genes were found to be misregulated following *Dach1* forced pan-neuronal expression. 404 and 210 of them showed reduced or increased expression, respectively. As expected, genes encoding for Prdm12, TrkA, TrkB or TrkC were not deregulated (Fig. 5C).

Analysis of the putative endogenous expression of the 40 most upregulated and 40 most downregulated genes in adult somatosensory neurons revealed that these genes are differentially enriched (Data ref: Kupari and Ernfors, 2023). Many of the misregulated genes were characteristic of nociceptors, suggesting that this population is predominantly impaired following Dach1 forced pan-neuronal expression, while genes found in the endogenous Dach1 compartment were only found among upregulated genes, suggesting that ectopic expression of Dach1 results in ectopic expression of genes characteristic of its endogenous expression domain (Fig. 5D). Gene ontology

analysis of the 100 most upregulated and 100 most downregulated genes further revealed that they were enriched in terms correlating with ion channels or transporters, reflecting their involvement in the functional components of somatosensory neurons (Fig. 5E). Likewise, several of these misregulated genes were previously reported to have a physiological function in the activity of somatosensory neurons (*TrpM8, Scn10a, Mrgpra1, Oprk1*) (Dhaka et al, 2007; Colburn et al, 2007; Bautista et al, 2007; Faber et al, 2012; Liu et al, 2009; Snyder et al, 2018), or to be markers of defined somatosensory subpopulations (*Th, MrgprD, MafA, Calca*) (Sharma et al, 2020; Bourane et al, 2009). Upregulation of *Th* and *Mef2C* and downregulation of *Calca*/CGRP, *MafA, TrpM8* and *Cdh3* were further validated by immunostaining or in situ hybridization on E18.5 transverse sections (Fig. 5F,G). Moreover, analysis of TH and Mef2C expression at E13.5 did not reveal any change, indicating that their observed expansion occurs as expected during the period of forced overexpression of Dach1 (Fig. 5F). These results indicate that the sustained pan-sensory expression of Dach1 results in an alteration of the transcriptional profile of somatosensory subtypes in which Dach1 is expected to become repressed.

## Maintenance of Dach1 in nociceptors alters pain-related behaviour

The misregulated transcriptional profiles observed following the forced pan-sensory expression of Dach1 led us to hypothesize that the observed transcriptional changes could have repercussions on the physiological capacities of somatosensory neurons. Unfortunately, the postnatal death of *AD1* OE mice precluded further analysis of such somesthetic features. To overcome this obstacle, we seeked to cross *Rosa26$^{lox-stop-lox-Dach1-IRES-EGFP/+}$* with a Cre-reporter mouse line allowing a narrower overexpression of Dach1 compared to the *Advillin$^{Cre}$* mouse line. We used therefore the *Scn10a$^{Cre/+}$* mice to generate *Scn10a$^{Cre/+}$; Rosa26$^{lox-stop-lox-Dach1-IRES-EGFP/+}$* (*SD1* OE) mice. *Scn10a* encodes for the nociceptor-specific sodium channel Nav1.8 which is expressed from E14.5 in small diameter somatosensory neurons and thus allows the sustained expression of Dach1 in nociceptors (Stirling et al, 2005). *SD1* OE mice were viable and fertile and showed wide expression of Dach1 in DRG (Fig. 5H). Analysis of glabrous skin free nerve endings suggested that innervation by nociceptive fibres is not altered in *SD1* OE mice (Fig. EV3A). However, analysis of dorsal horn innervation by TrkA$^+$ peptidergic nociceptive fibres revealed an abnormal expansion of the dorsal horn region receiving these afferents in *SD1* OE mice (Fig. EV3B). Nociception of *SD1* OE mice was then evaluated using the formalin test and showed that while the nocifensive response during the early phase is not markedly altered compared to controls, the response associated with the late phase is significantly reduced (Fig. 5I,J). We therefore conclude that the sustained expression of Dach1 in nociceptive neurons is not compatible with proper nociceptor function.

# Discussion

## Dach1 as a new transcription factor required for appropriate behavioural reactivity to touch

Here, we report the previously unrecognized function of Dach1 in the developing somatosensory system. Dach1 is a transcription

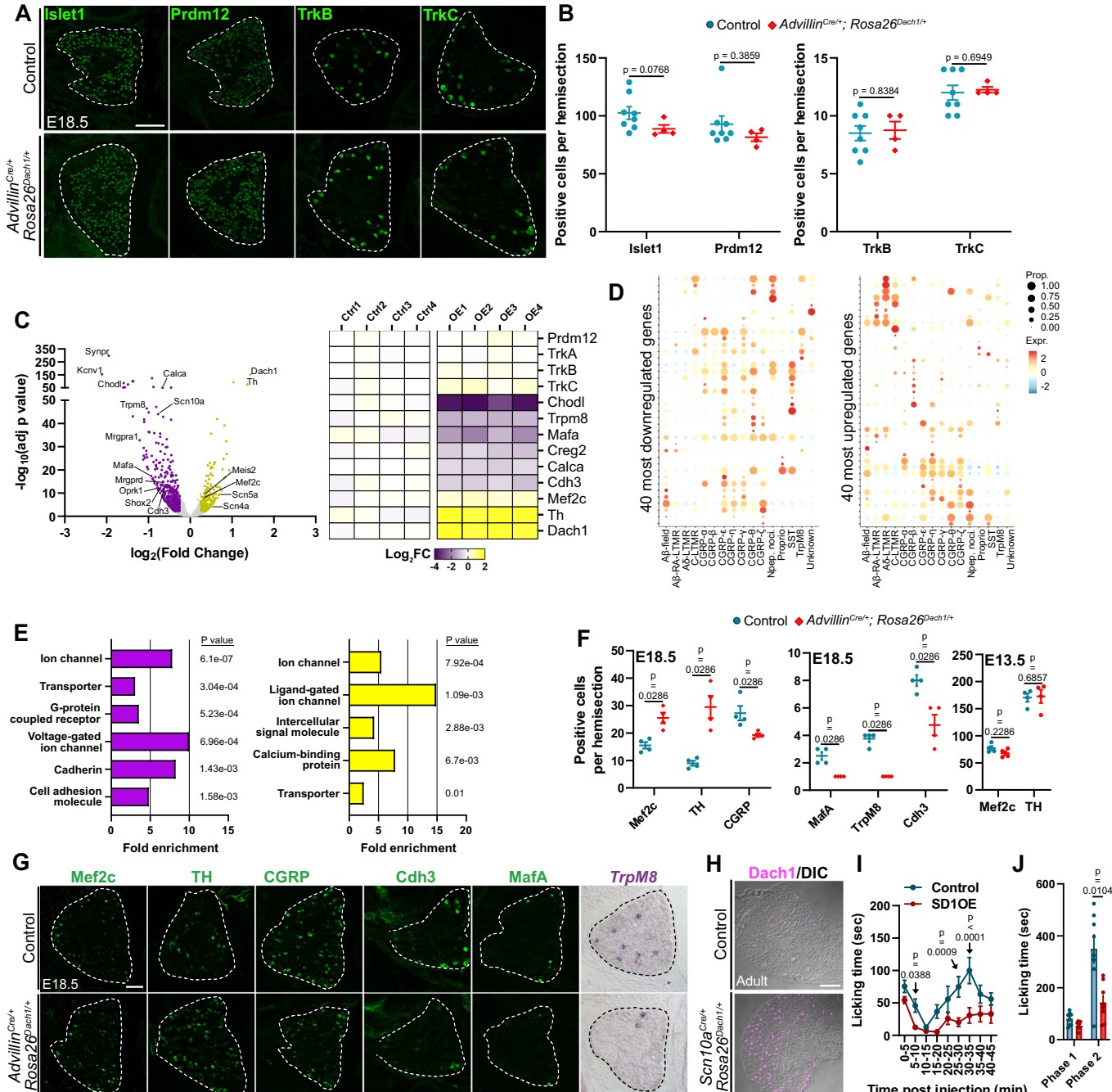

factor whose function has been extensively studied in other contexts and biological systems, notably in cancers and coronary angiogenesis (Raftrey et al, 2021; Chang et al, 2017; Zhu et al, 2023; Stewen et al, 2024). Our results indicate that Dach1 is detected from developing to mature post-mitotic somatosensory neurons. Initially pan-neuronal in developing DRG neurons, Dach1 becomes progressively turned off in most of these neurons to remain enriched in TrkB⁺ LTMR. In these neurons, Dach1 expression is required for appropriate behavioural reactivity to tactile stimulations. No alterations of mechanosensory fibres or of classical markers of LTMR subtypes are associated with embryonic Dach1

knock-out, suggesting that Dach1 is not involved in axonal growth or subtype specification. However, among the small set of genes downregulated following its depletion were genes directly involved in electrophysiological function (*Calb2, Scn5a, Kcnj12, Tmem266*). Also, genes overexpressed included several candidates encoding for cell-adhesion molecules (*Chodl, Pcdh7, Pcdh11x*) that could also affect the function of LTMR by modulating their peripheral and central ultrastructural features (Handler et al, 2023; Meltzer et al, 2023). Two of the upregulated genes, *Chodl* and *Shox2*, were found to be already co-expressed with *Dach1* in developing LTMR. The increase in their expression in the absence of Dach1 seems to have a

**Figure 5. Countered temporal refinement of Dach1 in developing somatosensory neurons results in wide transcriptional defects and altered nociception.**

(A) Representative pictures of immunostaining targeting indicated somatosensory subtype markers performed on thoracic transverse DRG sections of embryos of indicated genotypes. Note that the picture depicting Islet1 staining in $Advillin^{Cre/+}$; $Rosa26^{Dach1/+}$ condition is also used to illustrate data in Appendix Fig. S4D. (B) Quantification analysis comparing the mean number of neurons immunostained for indicated markers on transverse hemisections of E18.5 of indicated genotypes. Graphical data are presented as a scatter plot reporting mean ± SEM where each dot represents the mean value of one biological replicate (Control, $n = 8$; $Advillin^{Cre/+}$; $Rosa26^{Dach1/+}$, $n = 4$). Mann–Whitney tests. (C) Left, volcano plot showing the $-\log_{10}$(adjusted $p$-value) as a function of $\log_2$(Fold Change) of deregulated genes between DRG of E18.5 control ($n = 4$) and $Advillin^{Cre/+}$; $Rosa26^{lox-stop-lox-Dach1-IRES-EGFP/+}$ ($AD1$ OE; $n = 4$) embryos. Genes were identified by bulk RNA-seq and selected based on an adjusted $p$-value < 0.005 and a $\log_2$(Fold Change) > |±0.25|. Genes over this threshold which show downregulation in $AD1$ OE are labelled in purple while the ones showing overexpression are labelled in yellow. Genes falling below the threshold are indicated in grey. Right, Heatmap of select genes showing their equivalent or differential expression between control and $AD1$ OE conditions. (D) Bubble plot reporting the putative fraction of adult somatosensory subtypes that endogenously express the 40 most upregulated (left) or downregulated (right) genes identified in $AD1$ OE E18.5 DRG (Data ref: Kupari and Ernfors, 2023). (E) Gene ontology analysis reporting the protein class associated terms enriched among the 100 most upregulated (left) or downregulated (right) genes identified in $AD1$ OE E18.5 DRG. $P$ values were obtained using the Fisher's exact test. (F) Quantification analysis comparing the mean number of neurons stained for indicated markers on transverse hemisections of E18.5 or E13.5 embryos of indicated genotypes. Graphical data are presented as a scatter plot reporting mean ± SEM where each dot represents the mean value of one biological replicate (Control, $n = 4$; $Advillin^{Cre/+}$; $Rosa26^{Dach1/+}$, $n = 4$). Mann–Whitney tests. (G) Representative pictures of immunostainings with indicated antibodies and of in situ hybridization stainings using a probe targeting $TrpM8$ transcripts performed on E18.5 thoracic DRG transverse sections of embryos of indicated genotypes. (H) Representative pictures of adult DRG transverse sections from control $Scn10a^{Cre/+}$; $Rosa26^{lox-stop-lox-Dach1-IRES-EGFP/+}$ ($SD1$ OE) mice immunostained for Dach1 and acquired with a differential interference contrast (DIC). (I) Time course nocifensive response (licking time) of control ($n = 8$) and $SD1$ OE ($n = 8$) littermates until 45 min following formalin injection. Data are presented as the mean ± SEM. Two-way ANOVA with Fisher's LSD post hoc test. (J) Scatter dot plot showing the response of individuals of indicated genotypes during the first (0 to 5 min following injection) and the second (15 to 45 min following injection) phase of nocifensive behaviour of the formalin assay. Data are presented as a scatter plot reporting mean ± SEM where each dot represents the mean value of one biological replicate (Control, $n = 8$; $SD1$ OE, $n = 8$). Mann–Whitney tests. DRG in pictures are delineated by dashed lines. Scale bars, 100 μm. Source data are available online for this figure.

distinct cause. While cells with detectable *Chodl* transcripts are expanded among TrkB$^+$ neurons, an expansion of Shox2 is only detected at the protein level, suggesting that the upregulation of *Shox2* reported in this manuscript occurs in neurons in which it is already expressed. Therefore, the increase in Shox2$^+$ cells identified in Dach1 mutants may be the consequence of a higher abundance of *Shox2* transcripts in TrkB$^+$ LTMR neurons, which would be reflected by higher signal intensity and increased number of detectable immunostained nuclei.

## The broad-to-restricted expression of Dach1 associates with its function identified in TrkB$^+$ LTMR

Recent single-cell RNA-sequencing analyses of developing somatosensory neurons highlighted that neural crest cells biased to the somatosensory lineage first transition from an early immature post-mitotic neuronal state where several subtype-restricted transcription factors are initially expressed together. Among those are Runx1 and Runx3 which ultimately become mutually exclusive and are respectively involved in the specification of nociceptors and proprioceptors (Sharma et al, 2020). These broad-to-restricted transcription factors have in common that their depletion from their onset of expression results in defects restricted to the subtypes in which they are ultimately maintained. Here, we show that Dach1 behaves as a transcription factor with broad-to-restricted expression whose maintenance and extinction are both required to develop an unaltered sense of somesthesia. We report that depletion of *Dach1* during its broad or restricted expression period predominantly alters gene expression in developing TrkB$^+$ mechanosensory neurons by the end of embryogenesis. This observation suggests a discrepancy between the initiation of Dach1 expression in early post-mitotic somatosensory neurons and its functional turn-on when restricted to post-mitotic TrkB$^+$ neurons. Indeed, genes found misregulated by E18.5 following depletion of *Dach1* during its broad expression period are equally found misregulated following its depletion when enriched in TrkB$^+$ neurons. Despite with this strategy more general transient transcriptional defects cannot be ruled out during early

somatosensory neurogenesis following *Dach1* depletion during its broad expression period, these results suggest that Dach1 behaves as a broad-to-restricted transcription factor as defined by Sharma et al, 2020. These clues may indicate that a functional turn-on of Dach1 occurs following its expression refinement period in post-mitotic somatosensory neurons. Such temporal uncoupling between the expression of a transcription factor and the initiation of its functional activity was previously highlighted during the lineage specification in the trunk neural crest. Environment may play a predominant role in this discrepancy by conditioning the availability of critical cofactors or regulatory proteins (Soldatov et al, 2019). As such, Dach1 is found in dozens of cell types across the developing murine embryo (Qiu et al, 2022), suggesting that it may be engaged in multiple unrelated cell-type specification transcriptional programmes. Indeed, while early reports concluded that *Dach1* KO mice die at birth with no obvious abnormalities (Backman et al, 2003; Davis et al, 2001), subsequent analyses identifying a critical role for Dach1 in contexts as diverse as coronary artery development (Stewen et al, 2024; Chang et al, 2017; Raftrey et al, 2021), cortical somatostatin$^+$ long-range projection neurons development (Fisher et al, 2024) or mammary gland development (Jiao et al, 2019) have highlighted the functional complexity of its contribution to diverse developmental processes. In such various already elucidated and putative functional involvements, the temporal repression of Dach1 where it could be deleterious rather than its induction whenever needed may represent an optimized way allowing it to be rapidly engaged in highly unrelated developmental processes defined by their own chromatin regulatory landscapes and active gene regulatory networks.

## The broad-to-restricted expression of Dach1 is physiologically relevant to produce functional somatosensory neurons

The progressive compartmentalization of broad-to-restricted transcription factors within the subtypes in which they ultimately play a function poses the question of the biological relevance of this

progressive refinement. In other words, why do these transcription factors need to become subtype-specific if their reported early onset invalidation anyway results in subtype restricted defects? Opposite to broad-to-restricted transcription factors are transcription factors showing directly restricted late onset of expression. Among those is the ETS transcription factor Er81 whose neurotrophin-induced late post-mitotic expression is required for Ia proprioceptive neurons to synapse onto motor neurons in the spinal cord (Patel et al, 2003; Arber et al, 2000). Experimentally challenging Er81 late onset of expression by inducing the precocious initiation of ETS signalling in early post-mitotic somatosensory neurons previously demonstrated that Er81 temporal sequence of expression matters during neuronal development. A developmental switch appears to operate during somatosensory neurons development which results in precocious post-mitotic ETS signalling being detrimental while late onset ETS signalling being mandatory for neuronal development (Hippenmeyer et al, 2005). Here, we challenged a similar paradigm by countering the temporal refinement of Dach1 through a Dach1 overexpression mouse line allowing its conditional expression upon Cre-mediated activity (Raftrey et al, 2021). Through this strategy, we were able to sustain a pan-sensory expression of Dach1 during its late refined embryonic period or to express *Dach1* back into late post-mitotic *Scn10a*-expressing nociceptors. Forcing the maintenance of Dach1 broad expression resulted in wide transcriptional defects notably marked by the downregulation of a series of genes known to play functions in thermo/nociceptors (*TrpM8, Calca, Kcnv1, Oprk1, Scn10a*) and impaired the nocifensive behavioural response to formalin administration. These results may represent a requirement to compartmentalize the expression of Dach1 during the process of somatosensory neurogenesis. Dach1 may need to become subtype restricted when it finds itself in an environment where its expression would result in a functional activity detrimental for the developing sensory neuron (Appendix Fig. S5). This activity-dependent temporal refinement paradigm may serve as a basis to explain the expression dynamics of broad-to-restricted transcription factors.

## Molecular mechanisms involved in the broad-to-restricted expression of Dach1

What are the molecular cues resolving the selection of subtype-restricted transcription factors initially broadly expressed? It was previously shown that the NGF signalling pathway plays a critical role in refining the expression of several broad-to-restricted transcription factors, demonstrating that extrinsic cues are important for this refinement (Meltzer et al, 2021; Sharma et al, 2020). As Dach1 becomes progressively excluded from nociceptors, which represent an average 80% of DRG somatosensory neurons, we challenged whether the NGF signalling pathway is also required to resolve the subtype-restricted refinement of Dach1. However, we found that this mechanism does not apply to *Dach1*. The observation that Prdm12, a major determinant of nociceptor lineage, remains expressed in $NGF^{-/-};Bax^{-/-}$ DRG prompted us to hypothesize that it could support the temporal extinction of *Dach1* in developing nociceptors. Prdm12 is a transcriptional regulator putatively acting as an epigenetic repressor. Increasing evidence indicate that it plays temporal stepwise functions from the emergence of the nociceptive lineage to mature nociceptors in adult, functionally switching its transcriptional targets as development proceeds. In early born nociceptors, Prdm12 represses the out-of-context expression of visceral determinants *Phox2a/b* and is required

for nociceptor development and survival through the initiation and maintenance of the expression of *Ntrk1*/TrkA. In adult nociceptors, it modulates genes in a way that affects nociceptors homeostasis (Vermeiren et al, 2023; Desiderio et al, 2019; Landy et al, 2021; Latragna et al, 2022; Kokotović et al, 2021). Through deletion of Prdm12 in late post-mitotic nociceptors using $Scn10a^{Cre/+};Prdm12^{flox/flox}$ mice, we were able to examine the consequences in DRG of the loss of Prdm12 in conditions that did not result in *Phox2a/b* ectopic expression or in agenesis of the nociceptive lineage (Vermeiren et al, 2023). In the absence of Prdm12, Dach1 expression was expanded, indicating that Prdm12 contributes to its temporal refinement. Moreover, our ChIP experiments allowed us to identify a Prdm12-bound region in the Dach1 locus, suggesting that Dach1 is a direct transcriptional target of Prdm12. Intriguingly, the early pan-neuronal expression of Dach1 implies that in early born nociceptors, Dach1 and Prdm12 expression initially co-exist. This observation points out to a developmental switch where Prdm12 becomes competent to repress Dach1 at a time when its function is expected to become incompatible with the maturation of nociceptors. Prdm12 is a histone methyltransferases-related protein that failed to reveal de novo DNA binding motifs and does not carry intrinsic histone methyltransferase function (Thélie et al, 2015). These observations led to the hypothesis that Prdm12 may function with other transcription factors and cofactors (Nahorski et al, 2015). It has been suggested that it may in part function through the recruitment of the histone methyltransferase G9a, but mixed conclusions have been drawn from this initial hypothesis (Yang and Shinkai, 2013; Tsimpos et al, 2024). However, because of its putative function as a bridge to bring together proteins building transcriptional regulatory complexes, its function may deeply depend on the cellular context, which may account for the functional switch in its ability to repress *Dach1*. However, how Prdm12 is able to bind to the *Dach1* locus and to repress its expression currently remains unresolved at the mechanistic level.

In conclusion, through its maintenance in LTMR and extinction in nociceptors, Dach1 is critical for somatosensory neurons development. The temporal refinement of its expression appears to coincide with the triggering of its functional competence, hypothesized to be driven by a change in the cellular context. Overall, through the lens of *Dach1*, our study suggests that the broad-to-restricted temporal expression followed by several transcription factors is physiologically relevant to achieve appropriate development in somatosensory neurons.

## Methods

**Reagents and tools table**

| Reagent/Resource | Reference or Source | Identifier or Catalog Number |
|---|---|---|
| **Experimental models** | | |
| *Dach1^flox/flox* | Pr. Kristy Red-Horse (Chang et al, 2017) | N/A |
| *Rosa26^Dach1* | Pr. Kristy Red-Horse (Raftrey et al, 2021) | N/A |
| *Advillin^Cre* | (Zhou et al, 2010) | N/A |
| *Wnt1^Cre* | (Danielian et al, 1998) | N/A |

| Reagent/Resource | Reference or Source | Identifier or Catalog Number |
|---|---|---|
| *Scn10a*<sup>Cre</sup> | (Stirling et al, 2005) | N/A |
| *Prdm12*<sup>flox/flox</sup> | (Desiderio et al, 2019) | N/A |
| *Bax KO* | (Knudson et al, 1995) | JAX:002994 |
| *NGF KO* | (Crowley et al, 1994) | JAX:003312 |
| *Prdm12*<sup>KI</sup> | This study | N/A |
| **Recombinant DNA** | | |
| *TrpM8* in situ probe | Ren et al, 2014 | N/A |
| *Chodl* in situ probe (TOPO II vector) | This study | N/A |
| **Antibodies** | | |
| Rabbit anti-Dach1 | Proteintech | 10914-1-AP |
| Mouse anti-Islet1 | DSHB | 39.4D5 |
| Guinea pig anti-Prdm12 | Home made (Desiderio et al, 2019) | N/A |
| Goat anti-TrkB | R&D Systems | AF1494 |
| Goat anti-TrkC | R&D Systems | AF1404 |
| Goat anti-TrkA | R&D Systems | AF761 |
| Guinea pig anti-Parvalbumin | Synaptic Systems | 195 004 |
| Guinea pig anti-c-Maf | Gift from C. Birchmeier | N/A |
| Guinea pig anti-MafA | Gift from C. Birchmeier | N/A |
| Rabbit anti-TH | Abcam | Ab6211 |
| Sheep anti-TH | Novus Biologicals | NB300-110 |
| Goat anti-Ret | R&D Systems | AF482 |
| Goat anti-CGRP | Abcam | Ab36001 |
| Rabbit anti-Cleaved Caspase-3 | Cell Signaling | #9661 |
| Rabbit anti-Mef2C | Cell Signaling | D80C1 |
| Rabbit anti-S100β | Agilent | GA50461-2 |
| Chicken anti-NF200 | Abcam | Ab4680 |
| Rabbit anti-Calretinin | Millipore | AB5054 |
| Rabbit anti-Calretinin | Swant | 7697 |
| Mouse anti-Shox2 | Santa Cruz | Sc-81955 |
| Mouse anti-βIII-Tubulin | Covance | MMS-435 |
| Rat anti-BrdU | Abcam | Ab6326 |
| Rabbit anti-Homer1 | Synaptic Systems | 160 003 |
| Goat anti-rabbit Alexa Fluor 488 | Invitrogen | A11008 |
| Donkey anti-rabbit Alexa Fluor 488 | Abcam | Ab150073 |
| Donkey anti-mouse Alexa Fluor 594 | Invitrogen | A21203 |
| Goat anti-Guinea Pig Alexa Fluor 594 | Invitrogen | A11076 |
| Donkey anti-Guinea Pig Alexa Fluor 488 | Jackson Laboratories | 706-545-148 |
| Donkey anti-Goat Alexa 405 | Invitrogen | A21207 |
| **Oligonucleotides and other sequence-based reagents** | | |
| *Chodl* forward 5′-CTGGTTTGGAACATGCTGGGC-3′ | This study | N/A |
| *Chodl* reverse 5′-TTACTCCAAGATGTCTAAGTATACTGGTGG-3′ | This study | N/A |
| Cre – Fwd: 5′-CGATGCAACGAGTGATGAGGTTC-3′ | Desiderio et al, 2019 | N/A |
| Cre – Rev: 5′-GCACGTTCACCGGCATCAAC-3′ | Desiderio et al, 2019 | N/A |
| *Bax* WT – Fwd: 5′-GCAGAGGGTTAAAAGCAAGG-3′ | JAX Protocol | JAX:002994 |

| Reagent/Resource | Reference or Source | Identifier or Catalog Number |
|---|---|---|
| *Bax* Mutant– Fwd: 5′-CTTCCTGACTAGGGGAGGAG-3′ | JAX Protocol | JAX:002994 |
| *Bax* WT/Mutant– Rev: 5′-ACCCAGCCACCCTGGTCT-3′ | JAX Protocol | JAX:002994 |
| *Prdm12*<sup>flox</sup> – Fwd: 5′-GCTGATCGAGTCCAGGAGAC-3′ | Desiderio et al, 2019 | N/A |
| *Prdm12*<sup>flox</sup> – Rev: 5′-CCAAACATCCACAACCTTCA-3′ | Desiderio et al, 2019 | N/A |
| CSD-Dach1-ttR: 5′-AACAATTCTTGTCCTTCACGTGCCC-3′ | Chang et al, 2017 | N/A |
| CSD-Dach1-F: 5′-CTCCTGAAGATGAGGAGCTCACCCC-3′ | Chang et al, 2017 | N/A |
| *Rosa26*<sup>Dach1/+</sup> - 1: 5′-CTCTGCTGCCTCCTGGCTTCT-3′ | Raftrey et al, 2021 | N/A |
| *Rosa26*<sup>Dach1/+</sup> - 2: 5′-CGAGGCGGATCACAAGCAATA-3′ | Raftrey et al, 2021 | N/A |
| *Rosa26*<sup>Dach1/+</sup> - 3: 5′-TCAATGGGCGGGGGTCGTT-3′ | Raftrey et al, 2021 | N/A |
| *NGF* WT: 5′-CAGGCAGAACCGTACACAGA-3′ | JAX Protocol | JAX:003312 |
| *NGF* Mutant: 5′-CCTTCTATCGCCTTCTTGACG-3′ | JAX Protocol | JAX:003312 |
| *NGF* WT/Mutant: 5′-CTGTCACTCGGGCAGCTATT-3′ | JAX Protocol | JAX:003312 |
| *Colq* RNAScope probe | ACD Bio-techne | Cat# 496211 |
| *Dach1* RNAScope probe | ACD Bio-techne | Cat# 412071 |
| *Shox2* RNAScope probe | ACD Bio-techne | Cat# 579051 |
| *Chodl* RNAScope probe | ACD Bio-techne | Cat# 450211 |
| **Chemicals, Enzymes and other reagents** | | |
| BrdU | ThermoFischer Scientific | B23151 |
| **Software** | | |
| GraphPad Prism | GraphPad | https://www.graphpad.com |
| ImageJ | Schneider et al, 2012 | https://imagej.nih.gov/ij/ |
| **Other** | | |
| FlexAble Coralite ® Plus 488 antibody Labeling Kit | Proteintech | KFA041 |
| TSA Vivid fluorophore 520 | ACD Bio-techne | #323271 |
| TSA Vivid fluorophore 570 | ACD Bio-techne | #323272 |
| TSA Vivid fluorophore 650 | ACD Bio-techne | #323273 |

## Methods and protocols

### Mice

All mice were maintained on a C57Bl/6J background except *Dach1*<sup>flox/flox</sup> and *Rosa26*<sup>Dach1</sup> mice which were of mixed C57Bl/6J and CD1 backgrounds. Mice were provided *ad libitum* with standard mouse lab pellet food and water and housed at room temperature with a 12 h light/dark cycle. The experimental protocols were approved by the CEBEA (Comité d'éthique et du bien être animal) of the IBMM-ULB (agreement number LA1500474) and conformed to the European guidelines on the ethical care and use of animals. *Dach1*<sup>flox/flox</sup> (Chang et al, 2017) and *Rosa26*<sup>Dach1</sup> (Raftrey et al, 2021) mouse lines were provided by Pr. Kristy Red-Horse (Department of Biology, Stanford University, Stanford, USA). *Advillin*<sup>Cre</sup> (Zhou et al, 2010) and *Wnt1*<sup>Cre</sup> (Danielian et al, 1998) *mouse lines* were provided by Dr. P. Carrol and Dr. A. Pattyn (Institut des Neurosciences de Montpellier, Montpellier, France). The *Scn10a*<sup>Cre</sup>

(Stirling et al, 2005) mouse line was provided by Dr. Emmanuel Bourinet (Institut de Génomique Fonctionnelle, Montpellier, France). The *Prdm12^flox/flox* line was generated in our laboratory from a previous study (Desiderio et al, 2019). *Bax^+/−* (JAX:002994) (Knudson et al, 1995) and *NGF^+/−* (JAX:003312) (Crowley et al, 1994) mice were ordered through Jackson Laboratories. The *Dach1^flox* allele was detected by the primers forward 5′-CTCCT GAAGATGAGGAGCTCACCC-3′ and reverse 5′-AACAATTCTT GTCCTTCACGTGCCC-3′. The *Rosa26^Dach1* allele was detected by the primers 5′-CTCTGCTGCCTCCTGGCTTCT-3′, 5′-CGAGG CGGATCACAAGCAATA-3′ and 5′-TCAATGGGCGGGGGTC GTT-3′. The Cre allele was detected by the primers forward 5′-CGATGCAACGAGTGATGAGGTTC-3′ and reverse 5′-GCACG TTCACCGGCATCAAC-3′. The *NGF KO* allele was detected by the primers 5′-CAGGCAGAACCGTACACAGA-3′, 5′-CCTTCTATC GCCTTCTTGACG-3′ and 5′-CTGTCACTCGGGCAGCTATT-3′. The *Bax KO* allele was detected by the primers 5′-GCAGAGGGT-TAAAAGCAAGG-3′, 5′-CTTCCTGACTAGGGGAGGAG-3 and 5′-ACCCAGCCACCCTGGTCT-3′. The *Prdm12^flox* allele was detected by the primers forward 5′-GCTGATCGAGTCCAGGA-GAC-3′ and reverse 5′-CCAAACATCCACAACCTTCA-3′. The *Prdm12^KI* allele was detected by primers forward 5′-GGTATC-CACCCTTGTCTCCA-3′ and reverse 5′-CTTCAGCCCCTTGTT-GAATACG-3′.

### Tissue processing and fixation for immunofluorescence

Whole embryos (E10.5-E15.5), vertebral column (E16.5, E18.5), DRG (adult), hindpaws and back skin were dissected in ice cold PBS and fixed for 10 to 15 min in 4% Paraformaldehyde at 4 °C. Tissues were then rinsed with ice cold PBS and cryopreserved by overnight incubation in a solution of PBS containing 30% sucrose at 4 °C and subsequently embedded in 15% sucrose-7.5% gelatin. For embryos, transverse sections of 12 μm or 14 μm, 25 μm for hindpaws and 30 μm for back skin, were done using a cryostat (Leica) and sections were kept at −20 °C until use.

Slides were blocked in a solution of PBS with 0.1% Triton X-100 (PBST) containing 10% of donkey serum for at least two hours at room temperature, then incubated in primary antibodies diluted in blocking serum overnight at 4 °C. Slides were incubated in secondary antibodies diluted in PBST for two hours at room temperature. To reveal TrkB⁺ lanceolate endings in adult hairy skin, appropriate primary antibodies were instead incubated overnight at room temperature with no other modification to the protocol.

Primary antibodies used were: rabbit anti-Dach1 (1:500, Proteintech, 10914-1-AP), mouse anti-Islet1 (1:200, DSHB, 39.4D5), guinea pig anti-PRDM12 (1:1000, Desiderio et al, 2019), goat anti-TrkB (1:500, R&D System, AF1494), goat anti-TrkC (1:500, R&D System, AF1404), goat anti-TrkA (1:500, R&D System, AF761), guinea pig anti-Parvalbumin (1:500, Synaptic Systems, 195 004), guinea pig anti-c-Maf (1:10,000, gift from Carmen Birchmeier), guinea pig anti-MafA (1:10,000, gift from Carmen Birchmeier), rabbit anti-Th (1:250, Abcam, ab6211), sheep anti-Th (1:2000, Novus Biologicals, NB300-110), goat anti-Ret (1:50, R&D Systems, AF482), goat anti-calcitonin gene related peptide (CGRP) (1:200, Abcam, ab36001), rabbit anti-cleaved Caspase-3 (1:1000, Cell Signaling, #9661), rabbit anti-Mef2c (1:400, Cell Signaling, D80C1), rabbit anti-S100β (1:1000, Agilent, GA50461-2), chicken anti-NF200 (1:1000, Abcam, ab4680), rabbit

anti-Calretinin (Millipore, AB5054, 1:1000), rabbit anti-Calretinin (Swant, 7697, 1:2000), mouse anti-Shox2 (1:400, Santa Cruz, sc-81955), mouse anti-βIII-Tubulin (1:200, Covance, MMS-435), rat anti-BrdU (1:100, Abcam, ab6326), rabbit anti-Homer1 (1:500, Synaptic Systems, 160 003). Secondary antibodies used were: goat anti-rabbit Alexa Fluor 488 (1:1000, Invitrogen A11008), donkey anti-rabbit Alexa Fluor 488 (1:800, Abcam, ab150073), donkey anti-mouse Alexa Fluor 594 (1:1000, Invitrogen, A21203), goat anti-guinea pig Alexa Fluor 594 (Invitrogen, A11076), donkey anti-goat Alexa Fluor 594 (1:1000, Invitrogen, A11058), donkey anti-guinea pig Alexa Fluor 488 (1:800, Jackson Laboratories, 706-545-148) and donkey anti-goat Alexa 405 (1:1000, Invitrogen, A21207). Anti-Shox2 primary antibody was directly detected using FlexAble Coralite ® Plus 488 antibody Labeling Kit for Mouse IgG2a (Proteintech, KFA041).

Images collection and analysis were performed using a widefield fluorescence microscope Zeiss Axio Observer Z1, a laser-scanning confocal microscope Zeiss LSM 710 using the Zeiss Zen black microscopy software or a Stellaris 5 confocal microscope from Leica Microsystems. Image analysis was performed using ImageJ (Schneider et al, 2012). For expression profile experiments cell counts were conducted with at least 10 DRG sections analysed per animal with at least two biological replicates. For comparisons of controls with mutant embryos, 18 to 24 DRG sections were analysed with at least four biological replicates. Samples from at least one control and one mutant animal were systematically collected on a same glass slide to reduce variability along the subsequent experimental procedures.

### 5-Bromo-2′-Deoxyuridine (BrdU) injection

Pregnant females were injected intraperitoneally at 100 μl per kg with a solution of sterile PBS containing 1 mg of BrdU (TF Scientific, B23151) for 100 μl of PBS. After injection, mice were returned to their home cage for 20 min. Mice were then killed by cervical dislocation and E10.5 embryos were harvested on ice and rapidly processed as described above.

### Embryo processing and tissue fixation for in situ hybridization

Samples were fixed overnight in 4% Paraformaldehyde, rinsed with PBS and cryoprotected through overnight incubation in 30% sucrose at 4 °C. Tissues were embedded and cut as described above.

To process with the in situ hybridization, tissues were post-fixed in 4% PFA at room temperature for 20 min. After a 3 min-long proteinase K treatment, the slides were post-fixed in PFA 4% - 0.2% glutaraldehyde for 20 min. Tissues were then incubated in prehybridization solution (50% Formamide, 5X SSC, 50 μg/ml yeast extract, 1% SDS, 50 μg/ml heparin) for 2–3 h at 70 °C. Slides were incubated overnight at 70 °C with digoxigenin labelled riboprobes diluted in prehybridization solution. The next day slides were washed three times with a solution containing 50% Formamide, 5X SSC, 1% SDS), then three times with a solution containing 50% Formamide and 2X SSC. After a wash using TBS - 0.1% Triton X-100 (TBST), slides were blocked with 5% lamb serum diluted in TBST and incubated with an anti-digoxigenin antibody conjugated to Alkaline phosphatase (1/2000, Roche, 11093274910) overnight at 4 °C. Slides were incubated twice in Alkaline phosphatase buffer (100 mM Tris pH 9.5, 100 mM NaCl, 50 mM MgCl₂, 0.1% Tween 20) and incubated with the NBT and BCIP substrates (Roche) diluted in the same buffer. After PBS

washes and dehydration, the slides were mounted with Eukitt hardening mounting medium (Sigma).

The *Chodl probe was* cloned by PCR from *Mus musculus* adult cortex cDNA using the following primers: *Chodl* forward 5′-CTGGTTTGGAACATGCTGGGC-3′ and *Chodl* reverse 5′-TTACTCCAAGATGTCTAAGTATACTGGTGG-3′. The PCR product was then cloned into the TOPO II dual promoter vector (Promega) and the riboprobe synthesized (linearized with SpeI, transcribed with T7). The plasmid containing the *TrpM8* probe was described elsewhere (Ren et al, 2014).

Images collection was performed using a Zeiss Axioskop 2 equipped with a Axiocam 305 colour camera and the ZEN blue software. Image analysis was performed with Image J. For comparisons of controls with mutant embryos, 18 to 24 DRG sections were analysed with at least four biological replicates. Samples from at least one control and one mutant animal were systematically collected on a same glass slide to reduce variability along the subsequent experimental procedures.

### RNAScope

For RNAScope, fresh frozen individual DRGs from adult wild-type mice were embedded in OCT (VWR cat.# 361603E) and sectioned at 10 μm while E18.5 dorsal columns were sectioned at 20 μm. Sections were kept at −80 °C until use. RNAScope multiplex fluorescent in situ hybridization (version 2) was carried out following the manufacturer's protocol (Advanced Cell Diagnostics, ACD). Slides were removed from −80 °C and were immediately post-fixed in cold (4 °C) 4% paraformaldehyde for 15 min. The tissues were then dehydrated in 50% EtOH (5 min), 70% EtOH (5 min) and 100% EtOH (2 X 5 min) at room temperature. After air drying the slides, hydrophobic boundaries were drawn around each section with a hydrophobic pen. After a 10 min incubation with hydrogen peroxidase at room temperature, the tissues were subjected to a protease IV treatment (25 min for adult tissues or 10 min for embryonic tissues). Dach1-C1 (ACD Bio-techne, cat# 412071), Colq-C2 (ACD Bio-techne, cat# 496211-C2), Chodl-C2 (ACD Bio-techne, cat# 450211-C2) and Shox2-C3 (ACD Bio-techne, cat# 579051-C3) probes were added to the slides and hybridized for 2 h at 40 °C For adult DRG signal was developed in channel 1 with TSA Vivid Dye 570 (1/1500, #323272, Bio-techne) and channel 2 with TSA Vivid Dye 520 (1/1500, #323271, Bio-techne). For E18.5 vertebral columns, signal was developed in channel 1 with TSA Vivid Dye 520 (1/1500), in channel 2 with TSA Vivid Dye 570 (1/1500) and in channel 3 with TSA Vivid Dye 650 (1/1500, #323273, Bio-techne). Slides were counterstained with DAPI and mounted with Dako Fluorescent Mounting Medium. Images were taken on a Zeiss LSM 710 confocal microscope for WT DRGs, on a Leica Stellaris 8 STED confocal microscope for E18.5 DRGs and analysed with Image J.

For RNAScope coupled to Immunofluorescence, RNAScope was carried out as described above, but DAPI counterstaining was omitted. Instead, after developing signal in channel 3, Immunofluorescence was carried out as described above, with a primary goat anti-TrkB (1/500, R&D Systems, AF1494) antibody and a secondary anti-goat Alexa488 (1/1000, Invitrogen, A11055). Images were taken on a Stellaris 5 confocal microscope and analysed with Image J.

### Generation of the Prdm12^KI mouse line

The *Prdm12* knock-in (*Prdm12^KI*) mouse line was generated through a CRIPSR-Cas9 mediated homology directed repair (HDR) strategy. A crRNA was designed using CRISPRdirect (http://crispr.dbcls.jp/) with the intent to produce a double strand break in the fifth exon of Prdm12, in 5′ from the STOP codon. 2.4 pmol/μl of annealed crRNA (10 nM, Integrated DNA Technologies, 5′-GGCGCTCACAGCACCATGGC-3′) and the tracrRNA were injected with 200 ng/μl of recombinant Cas9 protein (Integrated DNA Technologies) and 10 ng/μl of a single-stranded DNA template (Megamer® single-stranded DNA fragment, GenScript) into the pronucleus of B6D2F2 zygotes which were subsequently transferred to pseudopregnant CD1 mice. The single-stranded DNA template was constituted (from 5′ to 3′) of: a *Prdm12 exon 5* homology arm (75 nucleotides), a GGTGGT Glycine linker sequence, three repeated Flag tag sequences, one V5 tag sequence terminated by a TGA stop codon, one Internal Ribosome Entry Site (IRES), one sequence encoding for a Venus fluorescent protein, one TGA stop codon followed by a 3′UTR homology arm (75 nucleotides). Founder mice having properly integrated the *Prdm12^KI* allele were screened by PCR and appropriate integration of the knockin-in region was validated by Sanger sequencing. Germline transmission of the selected *Prdm12^KI* founder mouse was then validated by PCR.

### Chromatin immunoprecipitation and data processing

The spinal cord with attached DRG of E12.5 embryos (10 embryos/condition) or isolated DRG of E13.5 (18 embryos/condition) were collected in ice cold PBS and fixed in 1% Formaldehyde (Thermo scientific) in Phosphate Buffer Saline (PBS, pH 7.5) for 15 min at room temperature in a shaking plate. The samples were quenched by adding a final concentration of 125 mM Glycine (Sigma-Aldrich) during 5 min at room temperature. Finally, the samples were washed twice in ice cold PBS and stored at −80 °C. Tissues were resuspended in 900 μL of SDS Lysis buffer (50 mM Tris pH 8.1, 10 mM EDTA, 1% SDS, cOmplete EDTA-free Protease Inhibitor Cocktail, Roche, #cat 05892791001) and sonicated for four cycles of 10 min (30 s ON 30 s OFF) at high setting using a Bioruptor Plus sonication device (Diagenode). The resultant fragmented chromatin was centrifuged at $10,000 \times g$ for 10 min and the supernatant was collected. The chromatin was diluted two times in Chromatin dilution buffer (16.7 mM Tris pH 8.1, 2 mM EDTA, 0.1% SDS (wt/vol), 1% Triton X-100 (vol/vol), 167 mM NaCl, cOmplete EDTA-free Protease Inhibitor Cocktail). 10% of the chromatin was de-crosslinked for input and eluted in a total volume of 50 μl. 9 μL of de-crosslinked inputs were run on an agarose gel to determine chromatin shearing efficiency. The chromatin was confirmed to be between 150 and 400 bp before proceeding to the chromatin immunoprecipitation. 800 μl of chromatin was mixed with 3200 μl of chromatin dilution buffer and 5 μg of rabbit polyclonal anti-V5 (Abcam, ab15828) or normal rabbit IgG (Merk, 12-370). The samples were incubated at 4 °C overnight. The next day, 30 μL of magnetic beads (Cell signaling, #cat 9906) were added and incubated in rotation at 4 °C during 2 h. Beads were collected using a magnetic stand and washed three times with low salt buffer (20 mM Tris pH 8.1, 2 mM EDTA, 1% SDS (wt/vol), 200 mM NaCl) for 5 min at 4 °C in rotation and one time with high salt buffer (20 mM Tris pH 8.1, 2 mM EDTA, 1% SDS (wt/vol), 0.5 M NaCl) for 5 min at 4 °C. The bound chromatin was subsequently eluted in 200 μL of elution buffer (1%SDS (wt/vol), 100 mM NaHCO₃) at 65 °C during 30 min. A magnetic stand was used to separate the beads and the supernatant containing chromatin was collected in a separate Eppendorf tube. Input and immunoprecipitated chromatin samples were brought to a final volume of 200 μL and were de-crosslinked and

the DNA was extracted using the following protocol. To each sample, a final concentration of 200 mM NaCl was added and incubated overnight at 65 °C. The next day, the samples were treated with RNAse (1 µL/sample) for 30 min at 37 °C. Followed by the addition of 8 µL of Tris HCl pH 6.5, 4 µL 0.5 M EDTA and 1 µL Proteinase K (20 mg/mL) for 1 h at 45 °C. Finally, the DNA was extracted using the High Pure PCR Product Purification kit (Sigma-Aldrich, REF 11732676001). The final chromatin was diluted in 50 µL of elution buffer.

For ChIP-qPCR, the input DNA was diluted 100 times and each immunoprecipitated chromatin sample was diluted 10 times. The Luna Universal qPCR Master Mix following manufacturer's instructions for qPCR, adding 5 µL of DNA per reaction. The qPCR was run on a StepOnePlus PCR instrument (Applied BioSystem). Fold enrichment was calculated as the percentage of input values. Primers forward 5′-AATGAGCCTTTGCTTCGCTT-3′ and reverse 5′-TTTTTCTCAGCCCATCTGCG-3′ were used to detect the intronic *Dach1* region bound by Prdm12. Primers forward 5′-GCCTCGCTGACCGAAACTAA-3′ and reverse 5′-CACCCTTGCCAGCTGTTTTG-3′ were used to detect the Prdm12-bound region close to *Dbx1*.

For ChIP-seq, the collected chromatin was sequenced using an Illumina Novaseq (BrightCore Platform, http://www.brightcore.be/). Paired-end reads were trimmed using TrimmomaticPE to remove Illumina universal adaptors. For data analysis, trimmed reads were mapped to mouse genome mm10 using Bowtie2 (Langmead and Salzberg, 2012) with default parameters for paired-end sequencing. Duplicate reads were removed with MarkDuplicates tools (Picard suite). Peaks were called with the callpeak tool from MACS2 package (Zhang et al, 2008) with following parameters: -f BAMPE -g mm -q 0.05 –nomodel –call-summits -B –SPMR. For visualisation, deeptools package was used to generate bigwig files (Ramírez et al, 2014). ChIP-seq data have been deposited at Gene Expression Omnibus (GEO) under accession number GSE281760.

### RNA sequencing and data processing
Dorsal root ganglia were dissected in cold RNAse Free PBS and stored at −80 °C in TRIZOL (ThermoFisher cat.# 15596026). RNA was extracted using the illustra RNAspin Mini RNA isolation kit (GE Healthcare cat.# 25-0500-70). RNA quality was assessed by measuring the RNA integrity number (RIN) using a Fragment Analyzer HS Total RNA Kit. Library preparation for RNA-Seq was performed using the Illumina Stranded mRNA Prep, Ligation and the Illumina RNA UD Indexes Set A, Ligation starting from 300 ng of total RNA. The size range of the final cDNA libraries was determined by applying the SS NGS Fragment 1- to 6000-bp Kit on the Fragment Analyzer (average 340 bp). Accurate quantification of cDNA libraries was performed by using the QuantiFluor™ dsDNA System (Promega). cDNA libraries were amplified and sequenced by using an S4 flow cell NovaSeq6000; 300 cycles, 25 Mio reads/sample from Illumina. Sequence images were transformed with BaseCaller Illumina software to BCL files and demultiplexed to fastq files with bcl2fastq v2.20.0.422. Sequencing quality was determined using FastQC v. 0.11.5 software (http://www.bioinformatics.babraham.ac.uk/projects/fastqc/).

Sequences were aligned to the *Mus musculus* GRCm39 genome using the STAR aligner software version 2.7.8a, allowing for 2 mismatches within 50 bases. Subsequently, read counting was performed using featureCounts version 2.0.1. Read counts were analysed in the R/Bioconductor environment version 4.1.0 (www.bioconductor.org) using the DESeq2 package version 1.32.0. Candidate genes were selected as those with a FDR-corrected $P$-value < 0.05 and absolute log2 fold-change >1. Genes were annotated using the M musculus GTF file mm10 version 105 used to quantify the reads within genes. RNA-seq data have been deposited at Gene Expression Omnibus (GEO) under accession numbers GSE270844 and GSE270845. Gene ontology analysis was performed using PANTHER (Mi et al, 2019) and selecting for terms associated with Protein Component.

### Analysis of published single-cell RNA-seq datasets
Single-cell RNA-seq data of mouse developing and postnatal DRG neurons were plotted as a force-directed layout using the published webtool provided with the publication by Sharma et al (2020) (https://kleintools.hms.harvard.edu/tools/springViewer_1_6_dev.html?datasets/Sharma2019/all). The bubbleplots reporting gene expression in adult DRG subtypes were generated using the published webtool developed by the Ernfors group (Data ref: Kupari and Ernfors, 2023) and leveraging data collected from Sharma et al (2020) and Zeisel et al (2018) (https://ernforsgroup.shinyapps.io/MouseDRGNeurons/).

### Wholemount Pacinian corpuscules immunostaining and quantification
Immunostaining of Pacinian corpuscules associated with the ulna was performed on dissected skinned forearms (radius and ulna) subsequently grossly trimmed from their muscles and connective tissue using a scalpel blade. To avoid damaging Pacinian corpuscules in the process, radius and ulna were kept together during the immunostaining procedure. Once the immunostaining was achieved the radius and ulna were separated and excess remaining surrounding tissue was carefully removed using forceps under an Olympus SZX16 binocular equipped with a X-Cite 120Q fluorescence illuminator. The wholemount immunostaining procedure was performed as described elsewhere for skin samples (Chang et al, 2014). The number of Pacinian corpuscules associated with the anterior half of the ulna were counted in four different individuals of each genotype. The area of Pacinian corpuscules was measured using ImageJ.

### Quantification of Meissner corpuscules density
Immunofluorescence with anti-S100β and NF200 antibodies was carried out on 25 µm sections of glabrous pedal pads from hindpaws to visualize Meissner corpuscules. The number of Meissner corpuscules in a section was counted and the area of the epidermis of the same section was traced and measured using Image J. Density of Meissner corpuscules was determined by dividing the number of corpuscules by the area of the epidermis.

### Quantification of hair follicles associated LTMR endings
Images of serial sections of back hairy skin were cryosectioned at 30 µm thickness. Quantification of the relative proportion of innervated hair follicles was performed using ImageJ by counting the number of hair shafts associated with NF200+ or TrkB+ endings on four independent hairy skin sections of 16 mm length. Counterstaining with S100β allowed visualisation of terminal Schwann cells associated with LTMR terminals to facilitate the quantification.

### Quantification of Merkel cells innervation

Merkel cells innervation was visualized through Cytokeratine K8 (CK8) and NF200 immunostaining in both hairy and glabrous skin on sections of 30 or 25 μm thickness, respectively. Confocal images were processed using ImageJ and CK8$^+$ Merkel cells were considered innervated if touched by a NF200$^+$ nerve fibre.

### Quantification of relative fluorescence level

Quantifications of the relative fluorescence level of Dach1 and Mef2C immunostainings were performed in ImageJ. By tracing a line crossing the major axis of individual nuclei and measuring the resulting mean grey value of fluorescence intensity of nucleus pixels crossing this line, the average Dach1$^+$ of Mef2C$^+$ signal intensity was manually calculated for every cell analysed. Analysed images were obtained using a laser-scanning confocal microscope Zeiss LSM 710 and the Zeiss Zen black microscopy software. Care was taken to maintain the exact same parameters of image acquisition for all experiments. Whenever appropriate, Superforst glass slides on which samples were collected always contained side-by-side serial sections of one control and one experimental condition to minimize variation during the subsequent immunostaining procedures.

### Behavioural assays

Unless otherwise specified, all tests were performed on mixed cohorts of males and females. All testing were performed blind to genotypes. Details about the sex of mice and their exact genotypes can be found in Dataset EV3.

**Von Frey** Mice were placed on an elevated wire mesh grid into PVC chambers. Before the test, mice were habituated to the device for 1 h for two consecutive days. On the testing day, mice were placed in the chamber 1 h before Von Frey filaments application. The test was performed as described in Neubarth et al (2020) and Neubarth et al (2020). During the test, withdrawal response following Von Frey filament application on the palm of the left hindpaw was measured. Starting with the lowest force, each filament ranging from 0.008 g to 1.4 g was applied 10 times in a row with a break of 30 s following the fifth application. During each application, bend filament was maintained for 4–5 s. The number of paw withdrawals for each filament was counted.

**Sticky tape** A 2 cm² of laboratory tape was placed on the upper-back skin of mice just before they were placed on an elevated wire mesh grid into PVC chambers. The number of tape-directed reactions was then counted during 5 min. Considered responses were body shaking like a 'wet dog', hindlimb scratching directed to the tape, trying to reach the tape with the snout and grooming of the neck with forepaws.

**Rotarod** Rotarod behavioural experiment was carried out over a 5-day period. Mice were trained on the Rotarod apparatus at a fixed speed of 10 rotations per minute (rpm) on the first day and tested for the next four days. On the days of the tests, the speed of the device was progressively increased from 10 to 40 rpm over a 5-min period. The latency to fall of each mouse was then recorded and the mean time was reported after three consecutive trials separated by a recovery period of 5 min.

**Formalin** Mice were habituated for 15 min on the day of the test in a PVC chamber with 3 mirrors on the walls and wire mesh grid on the floor. The dorsal part of the left hindpaw of the mouse was then injected with 10 μL of a 5% formalin solution and directly observed during a 45-min period. The licking time of the injected paw was recorded during 5 min intervals. Formalin assay including AD1 cKO animals was selectively performed on male cohorts while the assays including SP12 cKO or SD1 OE mice were performed on both males and females.

### Quantification and statistical analysis

Quantitative analyses were carried out on at least 3 independent animals of each genotype. All statistical analyses and the generation of graphs were performed using the GraphPad Prism software. Statistical tests as well as the number of cells or animals per group used in each experiment are described in the legends of the figures, EV figures and Appendix figures wherever appropriate. In case a dataset did not follow normal distribution, non-parametric statistical tests have been applied. In case a dataset did not contain enough biological replicates to be able to determine whether they follow a normal distribution or not, non-parametric tests have been also applied. No statistical methods have been used to determine sample sizes. Exact *p*-values are reported wherever appropriate.

## Data availability

The datasets produced in this study are available in the following databases: RNA-Seq data: Gene Expression Omnibus GSE270844. Gene Expression Omnibus GSE270845. ChIP-Seq data: Gene Expression Omnibus GSE281760.

The source data of this paper are collected in the following database record: biostudies:S-SCDT-10_1038-S44318-025-00427-y.

## Peer review information

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

## Acknowledgements

We thank members of the Bellefroid lab and Dr. Alexandre Pattyn for discussions and feedback about this manuscript. We thank Dr. Stanislas Goriely for providing help with bioinformatics analysis. We thank Dr. Pauline Cabochette for providing us with BrdU-injected E10.5 embryos. We are grateful to Dr. Maya Dannawi and Dr. Panagiotis Tsimpos for help with the PCR screening of the Prdm12 knock-in mouse line. We thank the staffs of the CMMI (Center for Microscopy and Molecular Imaging) microscopy facility and of the

IBMM-ULB Animal facility for their support. We are grateful to Dr. E. Bourinet and Dr. A. Pattyn for providing mouse lines and to Dr. Carmen Birchmeier for sharing MafA and c-Maf antibodies. This work was supported by grants from the FNRS (PDR T.0020.20 and T.0012.22) and by a grant from the International Brachet Foundation (IBS, project 22-3) to EJB. TS is supported by a grant from the Fondation Rose et Jean Hoguet and a ULB-UMons co-financed fellowship. SD is a FRS-FNRS postdoctoral fellow.

## Author contributions

**Tünde Szemes**: Formal analysis; Investigation; Visualization; Methodology. **Alba Sabaté San José**: Formal analysis; Investigation; Methodology. **Abdulkader Azouz**: Data curation; Formal analysis. **Maren Sitte**: Data curation; Formal analysis. **Gabriela Salinas**: Data curation; Formal analysis. **Younes Achouri**: Resources. **Sadia Kricha**: Investigation. **Laurence Ris**: Supervision; Funding acquisition. **Kristy Red-Horse**: Resources. **Eric J Bellefroid**: Conceptualization; Funding acquisition; Project administration; Writing—review and editing. **Simon Desiderio**: Conceptualization; Formal analysis; Supervision; Funding acquisition; Investigation; Visualization; Methodology; Writing—original draft; Project administration.

Source data underlying figure panels in this paper may have individual authorship assigned. Where available, figure panel/source data authorship is listed in the following database record: biostudies:S-SCDT-10_1038-S44318-025-00427-y.

## Disclosure and competing interests statement

The authors declare no competing interests.

# Expanded View Figures

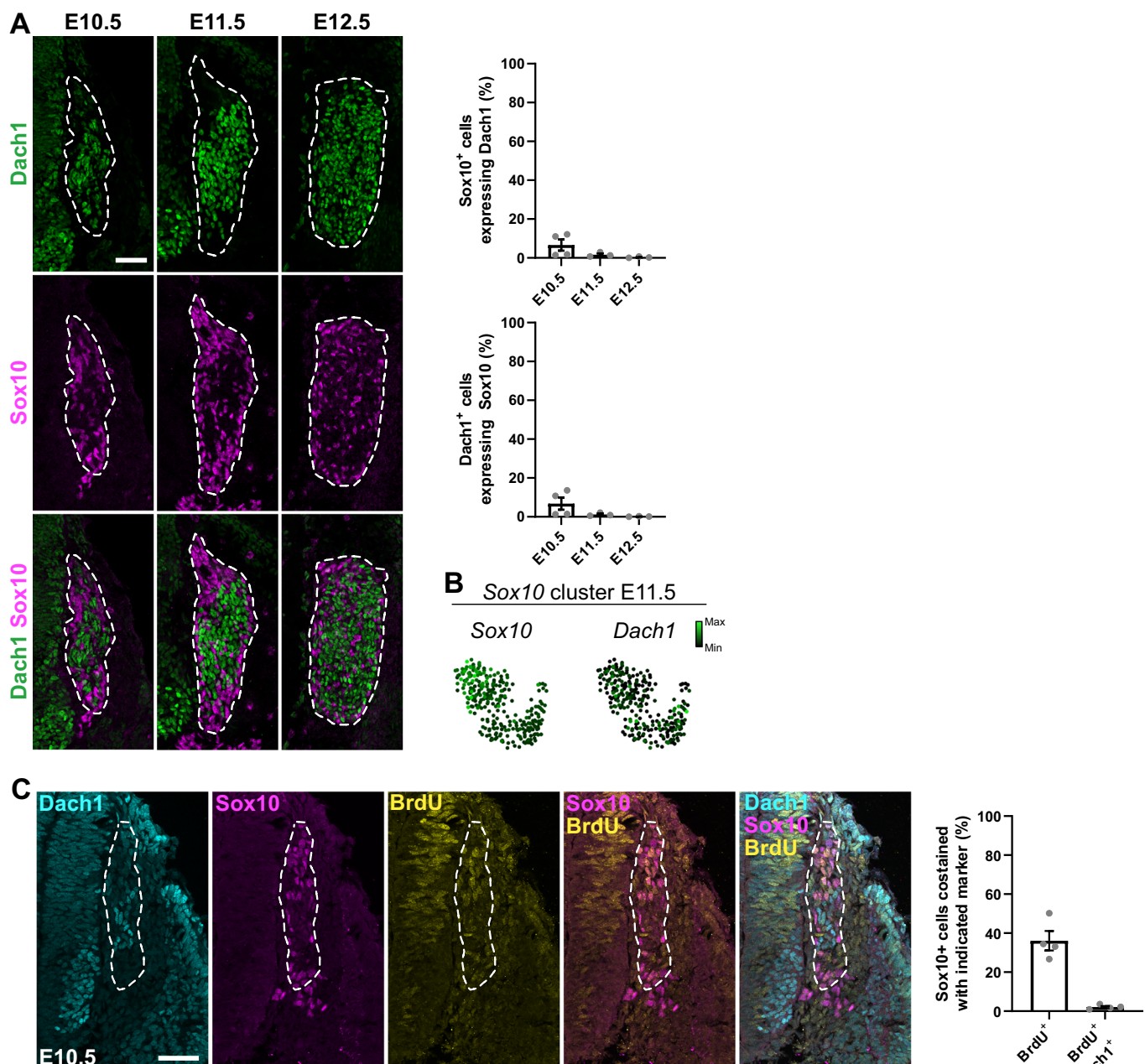

**Figure EV1.  Characterization of Dach1 expression in Sox10+ progenitors.**

(A) Left, representative pictures of double immunostainings of Dach1 with the progenitor marker Sox10 performed on thoracic DRG sections of murine embryos at the indicated developmental stages. Right, histogram quantifications of the percentage of Sox10+ cells expressing Dach1 or of Dach1+ cells expressing Sox10 at indicated developmental stages. Graphical data are presented as scatter plots reporting mean ± SEM where each dot represents the mean value of one biological replicate (E10.5, $n = 4$; E11.5, $n = 3$; E12.5, $n = 3$). (B) Force-directed layout of single-cell RNA-seq data of E11.5 DRG neurons found to express *Sox10* overlaid with expression levels of indicated genes. Data were obtained from Sharma et al, 2020. (C) Left, representative pictures of triple immunostainings of Dach1, Sox10 and BrdU performed on thoracic DRG sections of murine embryos at E10.5 following a 20 min BrdU treatment. Right, histogram quantifications of the percentage of Sox10+ cells expressing Dach1 or of Dach1+ cells expressing Sox10 at indicated developmental stages. Right, histogram quantification showing the percentage of Sox10+ cells co-labelled with BrdU alone or with BrdU and Dach1. Graphical data are presented as a scatter plot reporting mean ± SEM where each dot represents the mean value of one biological replicate ($n = 4$). DRG in pictures are delineated by dashed lines. Scale bars, 50 μm. Source data are available online for this figure.

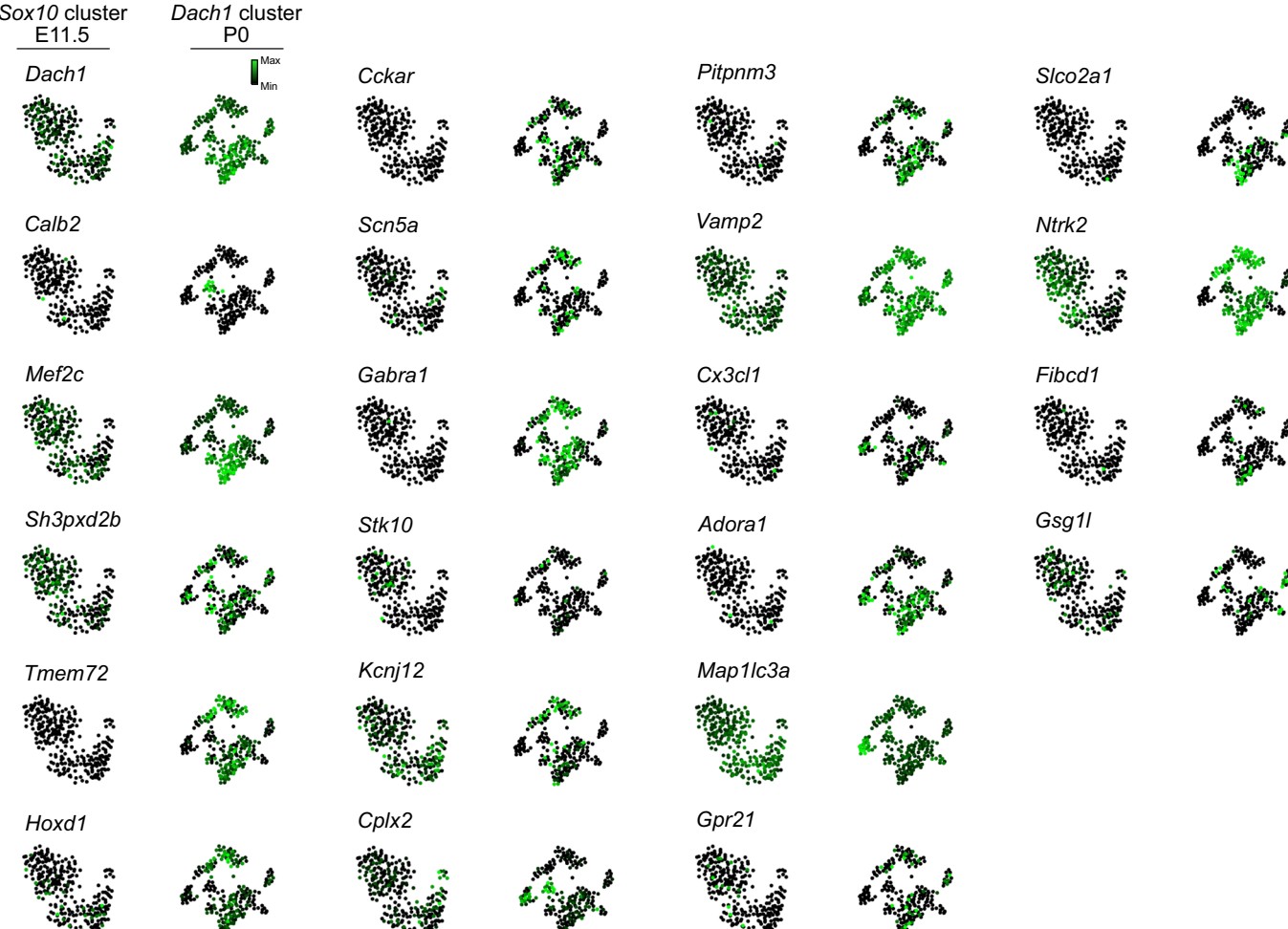

**Figure EV2.  Downregulated genes identified following Dach1 depletion are preferentially found in postmitotic neurons.**

Force-directed layout of single-cell RNA-seq data of E11.5 DRG neurons found to express *Sox10* and P0 DRG neurons found to express *Dach1* overlaid with expression levels of indicated genes. Data were obtained from Sharma et al, 2020.

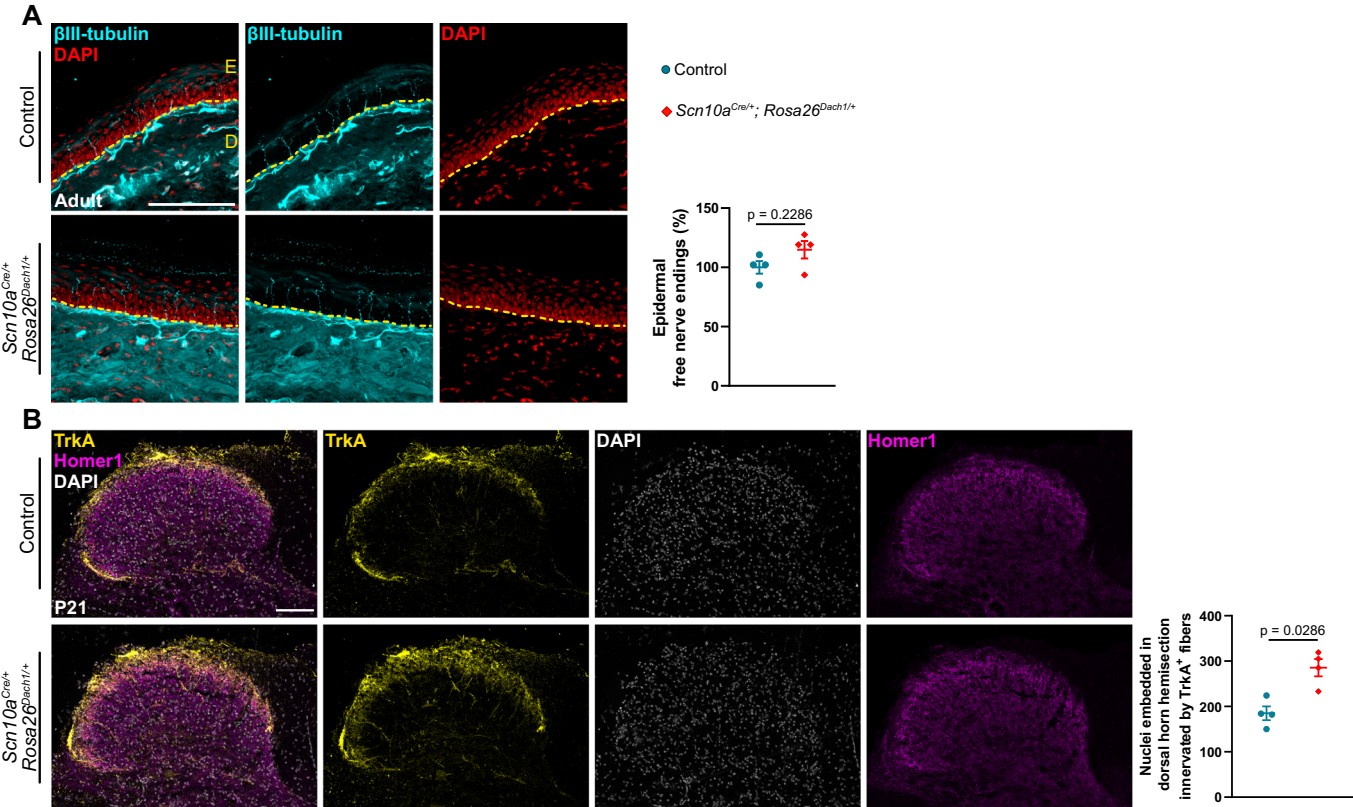

**Figure EV3. Characterization of central and peripheral innervation following Dach1 overexpression.**

(A) Left, representative pictures of immunostainings of βIII-Tubulin labelling nociceptive free nerve endings performed on glabrous hindpaw skin sections of control and *SD1* OE mice counterstained with DAPI (E, Epidermis; D, Dermis). The yellow dashed line delineates the epidermis/dermis boundary. Right, histogram quantification representing the percentage of nociceptive free nerve endings invading the epidermis in *SD1* OE mice ($n = 4$) compared to controls ($n = 4$). Graphical data are presented as a scatter plot reporting mean ± SEM where each dot represents the mean value of one biological replicate. Mann–Whitney test. (B) Left, representative pictures of immunostainings of TrkA and Homer1, respectively labelling nociceptive fibres and glutamatergic synapses performed on spinal dorsal horn section of P21 control ($n = 4$) and *SD1* OE ($n = 4$) mice and counterstained with DAPI. Right, histogram quantification representing the mean number of DAPI⁺ nuclei embedded in the dorsal horn region innervated by TrkA⁺ nociceptive fibres. Graphical data are presented as a scatter plot reporting mean ± SEM where each dot represents the mean value of one biological replicate. Mann–Whitney test. Scale bars, 100 µm. Source data are available online for this figure.

