## [Peer Review File · The EMBO Journal]

Temporal refinement of Dach1 expression contributes to the development of somatosensory neurons

Tünde Szemes, Alba Sabaté San José, Abdulkader Azouz, Maren Sitte, Gabriela Salinas-Riester, Younès Achouri, Sadia Kricha, Laurence Ris, Kristy Red-Horse, Eric Bellefroid, and Simon Desiderio

Corresponding author(s): Simon Desiderio (simon.desiderio@ulb.be) , Eric Bellefroid (eric.bellefroid@ulb.be)

Review Timeline:

Submission Date:	8th Aug 24
Editorial Decision:	5th Nov 24
Revision Received:	27th Jan 25
Editorial Decision:	20th Feb 25
Revision Received:	25th Feb 25
Accepted:	3rd Mar 25

Editor: Ioannis Papaioannou

Transaction Report:

Dear Simon,

Thank you again for the submission of your manuscript (EMBOJ-2024-118717) to The EMBO Journal for our consideration. As I have already informed you, your manuscript has been seen by three experts in the field, and we have received the full set of their comments (included below), which I have shared with you for your information. I would like to thank you for your detailed point-by-point response to these comments and your provisional revision plan, which were very helpful for us to reach a fair and balanced decision on the manuscript.

Although the referees agree that the findings are in principle interesting and potentially significant, referees #2 and #3 also identify several limitations in the study and the manuscript, and they raise a number of concerns. In particular, referee #2 points out that the possibility that Dach1 is expressed in different cell types and contexts, potentially with distinct functions in these different environments, could offer an alternative interpretation of the results that has not been considered. They also question whether the data suggesting biological significance of the "broad-to-restricted" expression pattern are sufficiently strong, while referee #3 also mentions that this model has not been investigated thoroughly enough. All referees list several more concerns and suggestions for strengthening the work and its conclusions further.

We acknowledge that in your detailed point-by-point response to the referees' concerns and comments, you provide further explanations to some of their comments, and you recognize that the majority of their points are valid expressing your willingness to address them by performing additional experimental work during a revision. In light of the referee reports and your thorough responses to them, as well as your provisional revision plan, which we find potentially sufficient to address the referees' concerns, I would like to invite you to submit a revised version of your manuscript along the lines you describe in your revision plan. I would like to kindly ask you to carefully consider all possible interpretations of the results, taking on board the comments of the referees, and discuss them in detail and in a balanced manner in your revised manuscript. Considering the mixed input we received from the referees, I would like to explain that the outcome of the re-review process cannot be guaranteed and depends on the completeness of your responses in this revised version, and the level of support we will receive from the referees upon re-review. Please also submit a detailed point-by-point response to the referees' comments along with your revised manuscript, describing all additions and changes to the manuscript.

I should add that it is The EMBO Journal policy to allow only a single round of major experimental revision, and acceptance of your manuscript will therefore depend on the completeness of your responses in this revised version. Please let me know if you have any questions or comments that you would like to discuss with me.

We generally allow three months as standard revision time (February 4th, 2025), but we may be able to grant an extension to allow enough time for the revision should the need arise. Should you foresee a problem in meeting the three-month deadline, please let us know. As a matter of policy, competing manuscripts published during this period will not negatively impact our assessment of the conceptual advance presented by your study. However, we request that you contact us as soon as possible upon publication of any related work, to discuss how to proceed.

Thank you again for the opportunity to consider your work for publication in The EMBO Journal. I am looking forward to your revision.

Best regards,

Ioannis

Instructions for preparing your revised manuscript

1. When you are ready to submit the revision, please upload:

- A Word file of the manuscript text (including legends of main Figures, EV Figures and Tables). Please make sure that changes are highlighted (or "tracked") to be clearly visible.

- Individual production-quality figure files (one file per figure). When assembling your figures, please refer to our figure

preparation guidelines in order to ensure proper formatting and readability in print as well as on screen:

If the data shown in a figure are obtained from n {less than or equal to} 2, please use scatter plots showing the individual data points.

- i. the name of the statistical test used to generate error bars and P values
- ii. the number (n) of independent experiments (please specify technical or biological replicates) underlying each data point (discussion of statistical methodology can be reported in the Materials and Methods section, but figure legends should contain a basic description of n , P , and the test applied)
- iii. the nature of the bars and error bars (s.d., s.e.m.).

- A point-by-point response to the referees' comments, with a detailed description of the changes made (as a word file). All referees' concerns must be fully addressed and their suggestions taken on board. When preparing your letter of response to the referees' comments, please bear in mind that this will form part of the Review Process File and will therefore be available online to the community. Please note that you have the possibility to opt out of the transparent process at any stage prior to publication by letting the editorial office know (contact@embojournal.org); if you do opt out, the Review Process File link will point to the following statement: "No Peer Review File is available with this article, as the authors have chosen not to make the review process public in this case.". For more details on our Transparent Editorial Process, please visit our website: <https://www.embopress.org/page/journal/14602075/authorguide#transparentprocess>

- Expanded View (EV) files (replacing Supplementary Information) that are collapsible/expandable online. A maximum of 5 EV Figures can be typeset. EV Figures should be cited as "Figure EV1, Figure EV2" etc. in the text, and their respective legends should be included in the manuscript file after the legends of regular figures. See detailed instructions regarding Expanded View files here:

- For the figures that you do NOT wish to display as Expanded View figures, they should be bundled together with their legends in a single PDF file called "Appendix", which should start with a short Table of Contents (including page numbers). Appendix figures should be referred to in the main text as: "Appendix Figure S1, Appendix Figure S2" etc. Please see detailed instructions here: <https://www.embopress.org/page/journal/14602075/authorguide#expandedview>

- A complete author checklist, which you can download from our author guidelines (<https://www.embopress.org/page/journal/14602075/authorguide>). Please note that the checklist will also be part of the Review Process File.

2. Please note that no statistics should be calculated and shown in Figures if $n=2$. Please also note that each p value should be reported as an exact value.

3. Before submitting your revision, primary datasets (and computer code, where appropriate) produced in this study need to be deposited in appropriate public databases (see <https://www.embopress.org/page/journal/14602075/authorguide#dataavailability>). In particular, we kindly ask you to deposit the ChIP-seq and RNA-seq datasets produced in your study. Their accession numbers, databases, and the specific URLs (links) should be listed in a formal "Data availability" section (placed after Methods), following the example:

"The datasets (and computer code, if applicable) produced in this study are available in the following databases:

RNA-Seq data: Gene Expression Omnibus GSE46843 (<https://www.ncbi.nlm.nih.gov/geo/query/acc.cgi?acc=GSE46843>)

Chip-Seq data: Gene Expression Omnibus GSE46748 (<https://www.ncbi.nlm.nih.gov/geo/query/acc.cgi?acc=GSE46748>)".

*** The Data Availability Section is restricted to new primary data that are part of this study. In case you have no data that require deposition in a public database, please state so instead of referring to the database: "Our study includes no data deposited in public repositories." under the heading "Data availability". ***

*** All links should resolve to a page where the data can be accessed. ***

*** Please remember to provide in the Data availability section of your revised manuscript reviewer passwords if the datasets are not yet public. ***

*** Please use detailed data citations for already available datasets that were re-analyzed in your study - for more information on the format, see point #9 below. ***

4. Please check that the title and the abstract of the manuscript are brief, yet explicit, even to non-specialists. The length of the title should not exceed 100 characters, and the abstract should be a single paragraph not exceeding 175 words.

5. All materials and methods need to be described in the manuscript using our "Structured Methods" format, which is now required for all research articles. According to this format, the Methods section includes a single "Reagents and Tools Table" - listing key reagents, experimental models, software and relevant equipment including their sources and relevant identifiers- followed by a "Methods and Protocols" section describing the methods. Please download and fill our Reagents and Tools Table template (.docx), which you can find in our author guide: <https://www.embopress.org/page/journal/14602075/authorguide#structuredmethods>. When submitting your revised manuscript, please do not include the Reagents and Tools Table in the Methods section of the manuscript but upload it as a separate file choosing the file type "Reagent Table".
6. Please also note our reference format: <https://www.embopress.org/page/journal/14602075/authorguide#referencesformat>.
7. At EMBO Press we ask authors to provide source data for the main manuscript figures. Our source data coordinator will contact you to discuss which figure panels we would need source data for and will also provide you with helpful tips on how to upload and organize the files.
8. Please remember: digital image enhancement is acceptable practice, as long as it accurately represents the original data and conforms to community standards. If a figure has been subjected to significant electronic manipulation, this must be noted in the figure legend or in the "Materials and Methods" section. The editors reserve the right to request original versions of figures and the original images that were used to assemble the figure.
9. Our journal encourages inclusion of data citations in the reference list to directly cite datasets that were obtained from public databases. Data citations in the article text are distinct from normal bibliographical citations and should directly link to the database records from which the data can be accessed. In the main text, data citations are formatted as follows: "Data ref: Smith et al, 2001" or "Data ref: NCBI Sequence Read Archive PRJNA342805, 2017". In the Reference list, data citations must be labeled with "[DATASET]". A data reference must provide the database name, accession number/identifiers, and a resolvable link to the landing page from which the data can be accessed at the end of the reference. Further instructions are available at: <https://www.embopress.org/page/journal/14602075/authorguide#referencesformat>.
10. We request authors to consider both actual and perceived competing interests. Please review our policy (<https://www.embopress.org/page/journal/14602075/authorguide#conflictsinterest>) and update your competing interests statement if necessary. Please name this section 'Disclosure and competing interests statement' and place it after the Acknowledgements section.
11. Please note that all corresponding authors are required to provide an ORCID ID upon submission of a revised manuscript (<https://orcid.org/>). Please find instructions on how to link your ORCID ID to your account in our manuscript tracking system in our Author guidelines (<https://www.embopress.org/page/journal/14602075/authorguide#authorshipguidelines>).
12. We use CRediT to specify the contributions of each author in the journal submission system. CRediT replaces the author contribution section, which should be removed from the manuscript. Please use the free text box to provide more detailed descriptions. See also guide to authors: <https://www.embopress.org/page/journal/14602075/authorguide#authorshipguidelines>.
13. Further information is available in our Guide For Authors: <https://www.embopress.org/page/journal/14602075/authorguide>
14. We would also welcome the submission of cover suggestions or motifs to be used by our Graphics Illustrator in designing a cover.
15. Please use the link below to submit your revision:
<https://emboj.msubmit.net/cgi-bin/main.plex>

Referee #1:

Szemes et al. study the role of the transcriptional repressor Dach1 in somatosensory neuron development. They demonstrate that this transcription factor is initially broadly repressed in DRG neurons and becomes later restricted to TrkB-positive A-LTMR neurons in a Prdm12-dependent manner. Analysis of the effects of Dach1 loss-of function revealed its requirement for the specification of the TrkB+ A-LTMR transcriptional program, while ectopic expression interferes with the transcriptional

maturation of other somatosensory types, most notably nociceptors. Consistent with these observations, behavioural paradigms were used to demonstrate reduced tactile sensitivity upon Dach1 loss and altered nociceptive behaviors upon ectopic Dach1 expression in nociceptors.

Taken together, this is a very well executed study that I thoroughly enjoyed reading. The experiments are well executed, the data is clearly presented and the paper is well written. I therefore only have few very minor comments that the authors will undoubtedly be able to resolve.

- 1) The following sentence in the summary section seems to be missing a word. "The relevance of this temporal refinement remains however unclear, these broad-to-restricted transcription factors being selectively involved in neurons in which they are ultimately maintained."
- 2) For the quantification shown in Figure 1 (or at least in Figure 1E), the authors only report an $n = 2$.
- 3) In Figure 1E the authors report that above 80% of TrkB+ neurons express Dach1. Moreover, the RNAseq analysis suggests that gain and loss of Dach1 have very similar effects on expression levels of Shox2 and Chodl, which are both endogenously found only in a small subset of TrkB+ neurons. Based on these observations I am wondering whether Shox2 and Chodl are expressed in the same neuronal subset and how their expression in the wild-type condition relates to Dach1 expression? Are Shox2 and Chodl potentially expressed in the TrkB+ neurons that endogenously do not express Dach1 and can the Dach1 gain- and loss-of function effects therefore be potentially explained by altered neuronal subtype ratios within the TrkB+ lineage under these experimental conditions?

Referee #2:

In this manuscript, the authors report that Dach1 is a transcription factor that exhibits broad expression in progenitors at E10.5 and early post-mitotic neurons, followed by restricted expression by E14.5 in a subset of somatosensory neurons-specifically, the low-threshold mechanoreceptor (LTMR) neurons. Most of the experiments aim to understand the functional significance of this "broad-to-restricted" expression pattern. However, an alternative interpretation of the data is that Dach1 is expressed in different cell types and contexts-such as progenitors, immature sensory neurons, and LTMR neurons-suggesting that it may play distinct roles in these different environments. This is not a mere question of semantics, as it has significant biological implications.

In considering whether the "broad-to-restricted" expression pattern has biological significance, perturbations that block this progression (such as *Scn10cre/+; Prdm12fl/fl; Wnt1Cre/+; Dach1flox/flox*, or *AdvillinCre/+; Dach1flox/flox* mice) would be expected to result in massive changes, likely comparable to the effects of Dach1 overexpression (using SD1 OE). However, this is not the case. Most of the observed phenotypes are relatively subtle, which suggests that Dach1 might play distinct roles in different contexts. Given that the phenotypes of *Wnt1Cre/+; Dach1flox/flox* and *AdvillinCre/+; Dach1flox/flox* mice are similar (Figure 4), this implies that removing Dach1 at E14.5 versus E12.5 has the most relevant effect. Therefore, the biological importance may lie in the expression of Dach1 itself, rather than in the broad-to-restricted pattern. In fact, no experiments conclusively support the claim that "the broad-to-restricted expression of Dach1 is physiologically relevant," so it would be important to adjust the conclusions accordingly.

Several key experiments could help strengthen the data presented in this paper:

1. In *Scn10cre/+; Prdm12fl/fl* mice, Dach1 expression is not restricted. What happens to the nociception of these animals when tested using the formalin assay?
2. The authors suggest there is a developmental switch in which "Prdm12 becomes competent to repress Dach1." What is the nature of this switch, and how does it occur?
3. The results in Figure 4E seem to contradict those in Figure 3A. The authors concluded that "depletion of Dach1 does not impact neurogenesis and does not alter the proportions of the main neuron subtypes." However, in Figure 4, it appears there are more Chodl+ and Shox2+ neurons. Clarification of this apparent discrepancy would be important.
4. It seems that the AD1 cKO mice largely phenocopy the W1D1 cKO mice (Figure 4), which suggests that the primary phenotype results from the loss of Dach1 at E14.5, rather than during the earlier broad expression period. Accurately reporting this result, instead of emphasizing the role of broad expression, would improve the manuscript.
5. The purpose of comparing gene expression in the dorsal root ganglion (DRG) at E18.5 between control and WD1 cKO mice is unclear, given that the AD1 cKO mice have a similar phenotype to the WD1 cKO mice in terms of cellular composition.
6. Overexpression of a transcription factor is expected to result in changes in gene expression, as Dach1 likely regulates many loci. The authors state that "pan-sensory expression of Dach1 is not compatible with the appropriate transcriptional maturation of somatosensory neurons." What exactly is meant by "transcriptional maturation"? Are there other ways to measure altered maturation, such as assessing firing or intrinsic electrophysiological properties? Alternatively, could it be that these neurons are not expanding their axons or forming synaptic connections at the appropriate time?

Minor comments:

- Line 343: "Figure 4G-H" is likely a typo, as there is no Figure 4G-H; it should reference Figure 5G-H.

Referee #3:

This work investigates the role of the transcription factor Dach1 in the functional maturation of somatosensory neurons. It shows how there is a temporal refinement of Dach1 expression, transitioning from broad to subtype-specific (in TrkB-expressing A-LTMR neurons) expression during neuronal development and show this is important for proper transcriptional maturation and functional specialization of this specific subtype of neuron. Although the authors thoroughly investigate the role of DACH1 in LTMR neurons and the effect of KO and OE on this subtype, the "de-coupling" of the function from broad to specific is not completely proven as progenitors are not investigated thoroughly and published single-cell data could be exploited to further address this point.

Major concerns:

1. The authors select DACH1 based on a single-study they report and show in Figure S1A the specific expression in LTMR neurons. They then characterize the expression of DACH1 in DRG sections in SOX10 progenitors. Since the single-cell paper has data from E11.5 and contain progenitors to unspecialized sensory neurons and then LTMR neurons at E12.5 it would be interesting to see the temporal dynamic expression of DACH1 from progenitors to neurons.
2. Furthermore, it would be interesting to see if by sub-subclustering the single-cell progenitors from E11.5 onwards DACH1 does not enrich in a specific progenitor population.
3. The authors show very low transcriptional changes between WD1 cKO and controls at E18.5. How many replicates did they use and what sequencing depth? Could batch effects contribute to this low levels of DEGs? How is the PCA for these samples?
4. The bulk analysis at E18.5 and the low number of DEGs could reflect the low proportion LTMR neurons at this stage, although the authors show that some genes are specific to the LTMR lineage by a dot plot a scoring method that shows how these genes specifically and significantly affect the LTMR lineage and not the progenitor stage could validate further their findings.
5. The authors state "Dach1 was already abolished by E12.5 in DRG of WD1204 cKO embryos, indicating that its depletion occurs as expected during its pan-sensory period" and they show staining with Isl1, however how many progenitors are present at this stage express DACH1?
6. The statement "Since the transcriptional alterations observed when Dach1 is depleted from the neural crest recapitulate those observed following its depletion in post-mitotic somatosensory neurons, they also suggest that the initiation of Dach1 expression is uncoupled from its functional turn-on" is not proven as the transcriptional changes in progenitors is not addressed and small changes in a subset of progenitors could affect later transcriptional changes.
7. The same holds true for overexpression, when identifying these signatures how do they change through the differentiation trajectory?

Minor:

1. Are the GOs shown in figure 5E significantly enriched?

Response to referees**Referee #1:**

Szemes et al. study the role of the transcriptional repressor Dach1 in somatosensory neuron development. They demonstrate that this transcription factor is initially broadly repressed in DRG neurons and becomes later restricted to TrkB-positive A-LTMR neurons in a Prdm12-dependent manner. Analysis of the effects of Dach1 loss-of function revealed its requirement for the specification of the TrkB+ A-LTMR transcriptional program, while ectopic expression interferes with the transcriptional maturation of other somatosensory types, most notably nociceptors. Consistent with these observations, behavioural paradigms were used to demonstrate reduced tactile sensitivity upon Dach1 loss and altered nociceptive behaviors upon ectopic Dach1 expression in nociceptors.

Taken together, this is a very well executed study that I thoroughly enjoyed reading. The experiments are well executed, the data is clearly presented and the paper is well written. I therefore only have few very minor comments that the authors will undoubtedly be able to resolve.

1) The following sentence in the summary section seems to be missing a word. "The relevance of this temporal refinement remains however unclear, these broad-to-restricted transcription factors being selectively involved in neurons in which they are ultimately maintained."

The sentence at lines 28 to 30 has been corrected and now reads "The relevance of this temporal refinement remains however unclear as these broad-to-restricted transcription factors are selectively involved in neurons in which they are ultimately maintained."

2) For the quantification shown in Figure 1 (or at least in Figure 1E), the authors only report an n = 2.

As the data in figure 1E are only descriptive we indeed did not perform the analysis on a number of biological replicates that is minimally required to assess statistical significance when comparing experimental conditions. However, to fulfil the remark of the reviewer, we now performed additional quantifications to add a biological replicate to the countings reported in figure 1E. We thus also updated the associated quantifications enumerated in the text (lines 140 to 144).

3) In Figure 1E the authors report that above 80% of TrkB+ neurons express Dach1. Moreover, the RNAseq analysis suggests that gain and loss of Dach1 have very similar effects on expression levels of Shox2 and Chodl, which are both endogenously found only in a small subset of TrkB+ neurons. Based on these observations I am wondering whether Shox2 and Chodl are expressed in the same neuronal subset and how their expression in the wild-type condition relates to Dach1 expression? Are Shox2 and Chodl potentially expressed in the TrkB+ neurons that endogenously do not express Dach1 and can the Dach1 gain- and loss-of function effects therefore be potentially explained by altered neuronal subtype ratios within the TrkB+ lineage under these experimental conditions?

As requested by the reviewer, we now performed quantifications on DRG E18.5 sections treated for RNAscope to reveal the expression of *Shox2* and *Chodl* inside and outside the *Dach1* compartment. The resulting panel and quantifications can be found in figure 4E and at lines 301 to 306 in the text. We observed that *Chodl*, which shows the most discrete expression pattern, is always co-expressed with *Shox2* when co-expressed with *Dach1* and that these triple-positive cells represent half of the *Chodl* population. Moreover, while *Shox2* at the protein level is only detected in a small fraction of *TrkB+* neurons (around 10% in Figure 4G, control condition), its expression among *TrkB+* neurons was found to be much wider when assessed by RNAscope (around 80%, figure 4H, control condition). This observation may come from a discrepancy regarding the efficiency of the immunolabelling to reveal detectable *Shox2*-positive nuclei compared to the higher sensitivity of the RNAscope. However, *Shox2* is known to undergo complex post-transcriptional regulation during somatosensory neuron development (Peng 2018, PMID: 30126905) and previous works by the team of Dr. Patrik Ernfors defining the percentage of *Shox2*-expressing cells at E12.5 in the *TrkB+* population identified an overlap of around 60% when *Shox2* is assessed at the RNA level (Abdo 2011, PMID: 22103411) and of around 25% when *Shox2* is assessed at the protein level (Peng 2018, PMID: 30126905), showing that we are not the first ones to observe this discrepancy. This is now highlighted and referenced in the text at line 313 to 317 and relate to the discrepancy observed between the RNAscope and immunolabelling results.

We also looked at the expansion of *Chodl* and *Shox2* in the *TrkB* population in the *WD1* cKO line (Figure 4H, lines 318 to 321). We observed that *Chodl* is expanded in *TrkB+/Shox2+* neurons. As *Chodl* and *Shox2* are already found to endogenously colocalize with *Dach1*, we cannot however conclude regarding a potential change in the balance of *TrkB* populations. Moreover, contrary to what is observed by immunostaining, the number of *Shox2+* cells is not affected following *Dach1* depletion when numbered by RNAscope, suggesting that the increase observed in our *WD1* cKO RNAseq dataset does not result from an expansion of *Shox2*-expressing cells in the absence of *Dach1* but is rather the consequence of an endogenous elevation of transcripts in cells already expressing *Shox2*. This discrepancy is now mentioned in the discussion section at lines 418 to 426.

Referee #2

In this manuscript, the authors report that *Dach1* is a transcription factor that exhibits broad expression in progenitors at E10.5 and early post-mitotic neurons, followed by restricted expression by E14.5 in a subset of somatosensory neurons-specifically, the low-threshold mechanoreceptor (LTMR) neurons. Most of the experiments aim to understand the functional significance of this "broad-to-restricted" expression pattern. However, an alternative interpretation of the data is that *Dach1* is expressed in different cell types and contexts-such as progenitors, immature sensory neurons, and LTMR neurons-suggesting that it may play distinct roles in these different environments. This is not a mere question of semantics, as it has significant biological implications.

Just like properly highlighted by the reviewer, we indeed started to investigate *Dach1* function with the initial hypothesis that *Dach1* may play different roles during its pan-neuronal period and during its LTMR-restricted period.

Our initial reasoning was that by depleting *Dach1* using the *Wnt1-Cre* line, which prevents its expression from the neural crest, *Dach1* would be abolished from its onset of expression during its broad expression period (between E10.5 and E12.5, when *Dach1* is also found in developing nociceptors as shown in Fig. 2A). In line with this, we show in the Appendix

Figure S2A that in the Wnt1-Cre; Dach1 cKO embryos, Dach1 is undetectable at E12.5 and probably even before.

Despite our expectation to observe dramatic effects following depletion of Dach1 during this broad period, no difference was found in dorsal root ganglia of Wnt1-Cre; Dach1 cKO compared to control in the total number of neurons (using pan-neuronal marker Islet1) by the end of embryogenesis (Fig. 3A). We inferred from this result that somatosensory neurogenesis is not compromised following depletion of Dach1 from the neural crest. We then looked at classical markers of the main somatosensory classes of neurons and again found no difference. This absence of dramatic phenotype prompted us to perform the RNA-seq described in figure 4 where we found that Dach1 depletion from its broad expression period onward results in defects selective of LTMR neurons. We interpret the absence of broad transcriptional defects as the proof that early onset depletion of Dach1 selectively affects a population of LTMR neurons. The fact that this phenotype is recapitulated by depleting Dach1 during its LTMR-specific period (using the Advillin-Cre; Dach1 cKO mouse line) further indicates that it predominantly functions during this restricted period.

An equivalent pipeline of analysis was used by Sharma et al (PMID: 31915380) that initially highlighted the concept of broad-to-restricted transcription factors in the context of somatosensory neurogenesis. In their paper, they first show that the transcription factors Pou4f2 and Pou4f3 are expressed broadly in somatosensory neurons before becoming subtype-restricted (extended Data Fig.5 and Fig. 3d-e of the Sharma study). They then demonstrate by transcriptomic analysis that the constitutive knock-out of these transcription factors (an equivalent strategy to our Wnt1-Cre mediated knock-out) selectively affects the subtypes in which these factors become ultimately restricted (Fig. 4 of the Sharma study). Based on these properties (selective subtype defects despite early broad expression), they introduce the concept of broad-to-restricted transcription factors in the context of somatosensory neuron development. In our paper we demonstrate that Dach1 behaves as such a broad-to-restricted transcriptional regulator because its depletion during its broad period, like the one of Pou4f2 and Pou4f3, selectively affects the subtypes in which it is ultimately maintained. The main difference at this stage of our study is that we further show that the misregulated genes are equally affected if we deplete Dach1 later, during its restricted period, which indeed points to a cell-type/context-dependent function of Dach1, with the exception that before it reaches this context, its expression is not associated with a significant function (no defects of neurogenesis, no broad/dramatic transcriptional changes in nociceptors). Such an absence of early function may at first glance appear counter-intuitive, however, it was previously demonstrated to occur in other developmental contexts, for example in the study by Soldatov et al., on the early development of the neural crest lineage (figure 2D-E of Soldatov et al, PMID: 31171666).

In considering whether the "broad-to-restricted" expression pattern has biological significance, perturbations that block this progression (such as Scn10cre/+; Prdm12fl/fl, Wnt1Cre/+; Dach1flox/flox, or AdvillinCre/+; Dach1flox/flox mice) would be expected to result in massive changes, likely comparable to the effects of Dach1 overexpression (using SD1 OE). However, this is not the case. Most of the observed phenotypes are relatively subtle, which suggests that Dach1 might play distinct roles in different contexts. Given that the phenotypes of Wnt1Cre/+; Dach1flox/flox and AdvillinCre/+; Dach1flox/flox mice are similar (Figure 4), this implies that removing Dach1 at E14.5 versus E12.5 has the most relevant effect. Therefore, the biological importance may lie in the expression of Dach1 itself, rather than in the broad-to-restricted pattern. In fact, no experiments conclusively support the claim that "the

broad-to-restricted expression of Dach1 is physiologically relevant," so it would be important to adjust the conclusions accordingly.

We unfortunately disagree with the referee regarding this interpretation of our data considering that Dach1 broad-to-restricted expression pattern is biologically significant, i.e. that Dach1 would initially play broad functions followed by LTMR-restricted functions. Our objective in this work was to understand why the expression of Dach1 (and by extension of other previously defined broad-to-restricted factors such as Pou4f2 and Pou4f3, which also play subtype selective roles despite their initial broad expression [Sharma et al, PMID: 31915380]) is shrunk as development proceeds, especially if it does not appear to play significant roles outside of the LTMR subtypes. Using gain-of-function mouse lines, we show that maintaining Dach1 when it must become restricted results in wide transcriptional defects, notably in the nociceptor compartment in which Dach1 is initially expressed, but later extinguished as shown in figure 2A. The observation that having Dach1 sustained in these cells (that previously expressed it during their developmental history but now should have downregulated it) results in defective nociceptors is the one that allows us to conclude that this broad-to-restricted phenotype is physiologically relevant. Not because it plays physiological functions during its broad and restricted periods, but instead because the temporal dynamics of its expression during the course of somatosensory neurons development is important for the appropriate development of neurons in which it must become extinguished by E14.5.

Several key experiments could help strengthen the data presented in this paper:
1. In Scn10cre/+; Prdm12fl/fl mice, Dach1 expression is not restricted. What happens to the nociception of these animals when tested using the formalin assay?

We now performed the experiment suggested by the reviewer and observed that, like what was observed following the gain-of-function of Dach1, Scn10acre/+; Prdm12fl/fl show reduced sensitivity to the formalin assay. These data are now presented in figure 2F-G and lines 189 to 192 in the manuscript.

2. The authors suggest there is a developmental switch in which "Prdm12 becomes competent to repress Dach1." What is the nature of this switch, and how does it occur?

We and others have studied Prdm12 function over the past and shown that it plays stepwise functions depending on the developmental stage and at adulthood in nociceptors (Bartesaghi et al., 2019; Desiderio et al., 2019; Kokotović et al., 2021; Landy et al., 2021; Latragna et al., 2022; Vermeiren et al., 2023). Despite the identification of a diverse array of phenotypes depending on the timing of Prdm12 depletion in developing dorsal root ganglia, we still do not know how it works. Prdm12 is a protein that might be recruited to DNA in association with other transcription factors and that has no apparent intrinsic enzymatic function (Yang and Shinkai, 2013; Thélie et al., 2015). The current view of how it works is that it may serve as a bridge to bring histone-modifying enzymes with DNA-binding proteins to regulate gene expression, as observed for other Prdm family members (Di Tullio et al., 2022; PMID: 33774927). Depending on the cellular context, these partners and other proteins being part of the transcriptional complexes in which Prdm12 is involved may change, which would account for its functional switch. Deciphering how Prdm12 works and is able to play diverse stepwise functions is currently one of the main focuses of our laboratory. The discussion has been modified to raise this point (see lines 519 to 527).

3. The results in Figure 4E seem to contradict those in Figure 3A. The authors concluded that "depletion of Dach1 does not impact neurogenesis and does not alter

the proportions of the main neuron subtypes." However, in Figure 4, it appears there are more Chodl+ and Shox2+ neurons. Clarification of this apparent discrepancy would be important.

The markers used in Fig. 3A are classical markers used in the field to quantify the general proportions of nociceptors, mechanoreceptors and proprioceptors, encompassing all the subtypes that these three cardinal populations contain. This is now specified at line 242. The purpose of these markers in Fig. 3A is to demonstrate that nociceptors, mechanoreceptors and proprioceptors are found in balanced proportions, in both cKO lines and in the control condition. To date, Chodl and Shox2 alone have not conclusively been used as *bona fide* markers of one of these three cardinal populations but may indeed be found together in a small subpopulation of TrkB+ neurons, as suggested by Referee #1. This has now been investigated in figures 4 E and 4H, which is developed in the response to Referee#1.

4. It seems that the AD1 cKO mice largely phenocopy the W1D1 cKO mice (Figure 4), which suggests that the primary phenotype results from the loss of Dach1 at E14.5, rather than during the earlier broad expression period. Accurately reporting this result, instead of emphasizing the role of broad expression, would improve the manuscript.

A sentence recapitulating the request of the reviewer has been added in the concluding paragraph of the results section describing the figure 4, at lines 321-323.

5. The purpose of comparing gene expression in the dorsal root ganglion (DRG) at E18.5 between control and WD1 cKO mice is unclear, given that the AD1 cKO mice have a similar phenotype to the WD1 cKO mice in terms of cellular composition.

The initial purpose of comparing gene expression in the dorsal root ganglia between control and WD1 cKO mice at E18.5 was driven by the absence of an apparent dramatic phenotype in this line. Indeed, we initially expected to observe dramatic defects in the WD1 cKO, notably in nociceptors that express Dach1 during its broad expression period, which did not turn out to be the case. This transcriptomic analysis at E18.5 (that in fact revealed that Dach1 depletion during its broad or restricted expression period results in equivalent outcomes, classifying Dach1 as a broad-to-restricted factor as defined by Sharma et al, PMID: 31915380) was carried out to determine whether we had missed any broad defects that would not be reflected by the numbers and proportions of neurons as assessed with molecular markers classically used in the field. We have now modified the text (lines 259 to 265) to emphasize this initial hypothesis, namely that we wanted to determine whether depletion of Dach1 using the WD1 cKO line would result in broad gene misregulations that we would not have noticed initially.

6. Overexpression of a transcription factor is expected to result in changes in gene expression, as Dach1 likely regulates many loci. The authors state that "pan-sensory expression of Dach1 is not compatible with the appropriate transcriptional maturation of somatosensory neurons." What exactly is meant by "transcriptional maturation"? Are there other ways to measure altered maturation, such as assessing firing or intrinsic electrophysiological properties? Alternatively, could it be that these neurons are not expanding their axons or forming synaptic connections at the appropriate time?

We mean here that there is a defective transcriptional maturation as several well documented functional components of these neurons such as ion channels and receptors (Nav1.8, TrpM8, MrgprD, ...) are found downregulated. This term may indeed not be the most appropriate in this context and we have now removed it from the updated version of the

manuscript. We now instead refer to altered transcriptional profile to describe the observed phenotype (lines 34-35, line 101, lines 327-328, line 378, line 381). We also analysed peripheral and central innervation by nociceptors in the glabrous skin and dorsal horn of SD1 OE mice. No changes were found in the periphery but altered central innervation was observed. These results are reported in Figure EV3 and described in the text at line 99 and lines 391 to 395.

Minor comments:

- Line 343: "Figure 4G-H" is likely a typo, as there is no Figure 4G-H; it should reference Figure 5G-H.

The typo has been corrected

Referee #3

This work investigates the role of the transcription factor Dach1 in the functional maturation of somatosensory neurons. It shows how there is a temporal refinement of Dach1 expression, transitioning from broad to subtype-specific (in TrkB-expressing A-LTMR neurons) expression during neuronal development and show this is important for proper transcriptional maturation and functional specialization of this specific subtype of neuron. Although the authors thoroughly investigate the role of DACH1 in LTMR neurons and the effect of KO and OE on this subtype, the "de-coupling" of the function from broad to specific is not completely proven as progenitors are not investigated thoroughly and published single-cell data could be exploited to further address this point.

Before answering point by point to the requests of referee #3, we would like to emphasize here that the comments of the referee prompted us to perform a formal characterization of Dach1 expression in somatosensory progenitors. This quantified analysis can be found in figure EV1 and replaces the original panel 1A which was not associated with quantitative measurements. Based on this new analysis, we conclude that Dach1 is not conclusively expressed in progenitors and have thus corrected this claim along the manuscript.

Major concerns:

1. The authors select DACH1 based on a single-study they report and show in Figure S1A the specific expression in LTMR neurons. They then characterize the expression of DACH1 in DRG sections in SOX10 progenitors. Since the single-cell paper has data from E11.5 and contain progenitors to unspecialized sensory neurons and then LTMR neurons at E12.5 it would be interesting to see the temporal dynamic expression of DACH1 from progenitors to neurons.

Following the suggestion of the reviewer, we isolated the cluster of Dach1 expressing cells in the study performed by Sharma et al (PMID: 31915380) at E11.5, E12.5 and E15.5. Comparison of Dach1 with Sox10 and Isl1 indicated that Dach1 is predominantly expressed in post-mitotic neurons, even at E11.5. Moreover, comparison of the Dach1/Ntrk1 subcluster (which are developing nociceptors) at E11.5, E12.5 and E15.5 indicates that nociceptors initially widely express Dach1 at E12.5 but are later massively excluded from the Dach1 cluster at E15.5. Of note, the small number of Ntrk1-expressing cells observed at E11.5 is in line with mechano- and proprioceptive neurons being predominant at this stage, while post-mitotic nociceptors are preferentially found from E12.5 onward (Lallemend & Ernfors, PMID: 22516617). These observations are represented in figure 1D and discussed in the text at lines 133 to 137.

2. Furthermore, it would be interesting to see if by sub-subclustering the single-cell progenitors from E11.5 onwards DACH1 does not enrich in a specific progenitor population.

We now provide an expanded view figure EV1 in which we characterize the expression of Dach1 from the point of view of progenitors. In this figure, we provide in the “B” panel a view of the Sox10 cluster isolated from E11.5 in the study by Sharma et al (PMID: 31915380) and highlight Dach1 expression in this cluster. We observe that Dach1 is not conclusively expressed in somatosensory progenitors, which is in line with the immunostaining experiments now performed in figure EV1A and EV1C. In figure EV1A, we analysed Dach1 expression compared to Sox10 at E10.5, E11.5 and E12.5 and identified that at E10.5, 6,6% of Sox10+ cells only expressed Dach1 at low levels. Moreover, figure EV1C shows the results of a 20' BrdU incorporation experiment at E10.5. Compared to the whole Sox10 population for which 36% of cells incorporate BrdU during this period, the proportion of Sox10+/Dach1+/BrdU+ cells is close to 0%. We thus conclude from these results that Dach1 is not conclusively expressed in progenitors as previously stated in our manuscript, a claim that has now been corrected throughout the text. We now suspect that the few Dach1+/Sox10+ cells initially observed may be the consequence of the initiation of Dach1 coupled to the dowregulation of Sox10 as progenitors exit the cell cycle, which have allowed the detection of Dach1 in some Sox10+ non-proliferating cells. These results are described in the manuscript at lines 116 to 128. We would like to deeply thank the reviewer for prompting us to carry a thorough analysis of Dach1 expression in the progenitor compartment, which allowed us to correct our initial claim.

3. The authors show very low transcriptional changes between WD1 cKO and controls at E18.5. How many replicates did they use and what sequencing depth? Could batch effects contribute to this low levels of DEGs? How is the PCA for these samples?

Five biological replicates were sequenced for control and WD1 cKO conditions. Based on PCA and clustering analysis, replicates 4 and 5 (for both control and cKO) behave differently from replicates 1, 2, and 3 due to collection of these groups of embryos on two different days. The sequencing was performed very deeply, with 26 to 47 million reads per sample, and all samples were processed together for library preparation and sequencing. Therefore, no batch effects are coming from the technical sequencing itself.

We have applied batch correction based on the experiment date and evaluated the replicates in terms of gene expression outcomes which led to a slightly higher number of genes found misregulated. Figures 4A, 4B, Appendix Table 1 and the text referring to it (lines 267 to 268) have been updated to take into account this new analysis. Finally, the use of a shrinkage step in DESeq2, which is a conservative correction on the fold changes to reduce the false positive findings, may be a reason for our small fold change.

4. The bulk analysis at E18.5 and the low number of DEGs could reflect the low proportion LTMR neurons at this stage. Although the authors show that some genes are specific to the LTMR lineage by a dot plot, a scoring method that shows how these genes specifically and significantly affect the LTMR lineage and not the progenitor stage could validate further their findings.

We leveraged the scRNA-seq Sharma study (PMID: 31915380) to provide layouts of the Sox10-expressing cells cluster at E11.5 compared to the Dach1-expressing cluster at P0 on which we have overlaid gene expression related to the significantly downregulated transcripts found in our bulk-RNAseq analysis (generated using the online tool provided with the
Sharma
paper,

https://kleintools.hms.harvard.edu/tools/springViewer_1_6_dev.html?datasets/Sharma2019/a II). These layouts, reported in the figure EV2 show that misregulated genes are mostly found to be robustly expressed at P0 and not in E11.5 Sox10+ cells. This figure is now referenced at lines 274 to 280 in the text.

5. The authors state "Dach1 was already abolished by E12.5 in DRG of WD1 cKO embryos, indicating that its depletion occurs as expected during its pan-sensory period" and they show staining with Isl1, however how many progenitors are present at this stage express DACH1?

We have not been able to detect endogenous Dach1 in Sox10+ progenitors at E12.5 (figure EV1A).

6. The statement "Since the transcriptional alterations observed when Dach1 is depleted from the neural crest recapitulate those observed following its depletion in post-mitotic somatosensory neurons, they also suggest that the initiation of Dach1 expression is uncoupled from its functional turn-on" is not proven as the transcriptional changes in progenitors is not addressed and small changes in a subset of progenitors could affect later transcriptional changes.

This claim has been removed from the new version of the manuscript as Dach1 is not conclusively expressed in NCC progenitors.

7. The same holds true for overexpression, when identifying these signatures how do they change through the differentiation trajectory?

We performed immunostaining of Mef2C and Th at E13.5 in AD1 OE dorsal root ganglia, which did not reveal increased expression contrary to what is observed at E18.5. This observation confirms that overexpression of these genes indeed occurs at the postmitotic stage (after E13.5) and that it is observed during the period of forced Dach1 pan-sensory expression in AD1 OE embryos. A histogram of this quantification was added in figure 5F and described at lines 375 to 377 in the text.

Minor: 1. Are the GOs shown in figure 5E significantly enriched?

We apologize as we did not take into account the significance of the enriched terms in our initial GO analysis. We now provide in figure 5E a new GO analysis reporting the enriched protein class associated terms and incorporating p-values. This new analysis led to an adaptation in the text at line 367.

Dear Simon,

Thank you for submitting your revised manuscript (EMBOJ-2024-118717R) to The EMBO Journal, and for your patience during re-review. Your manuscript has now been seen by two of the original referees that had previously assessed the initial version of your manuscript, and we have received their comments, which you can find below.

I am pleased to say that both referees find the revised manuscript significantly improved. Referee #3 mentions that all of their previously raised concerns have been fully and satisfactorily addressed, and now endorses this paper for publication. Referee #2 also recognizes that the manuscript is now improved, but also points out that two limitations of the study must either be addressed by further experimental work or acknowledged and discussed in the manuscript as appropriate. In addition, he/she identifies three examples of over-interpretation and suggests toning the respective claims down in the absence of conclusive experimental data.

We agree with the referee that the limitations of the study must be transparently discussed and that no over-statements or conclusions that are not fully supported by the available data can remain in the manuscript. Although no additional experimental work will be required to address them for the publication of the manuscript in The EMBO Journal, we kindly ask you to carefully rephrase all claims mentioned by the referee and make sure that the conclusions are not over-interpreted, in a final version of your manuscript. Any claims not fully supported by the available data should be phrased in a way that will clearly reflect their hypothetical/speculative nature. Please include in your resubmission a detailed point-by-point response to the referees' comments describing any changes to the manuscript.

From the editorial side, there are also a few corrections and changes that we need you to make in the final version of the manuscript before it can be accepted for publication. Please include in your resubmission a brief cover letter explaining how they are addressed:

- You can provide two more keywords (up to five can be listed in total) after the Abstract of your revised manuscript, if you wish.
- Thank you for depositing your RNA-seq and ChIP-seq datasets to appropriate databases and for providing the information in your Data availability statement. The reviewer tokens can now be removed from this statement. Please make sure that the datasets will be publicly available (from the provided URLs) at the time of publication.
- Our journal encourages inclusion of data citations in the reference list to directly cite datasets that were obtained from public databases. Data citations in the article text are distinct from normal bibliographical citations and should directly link to the database records from which the data can be accessed. In the main text, data citations are formatted as follows: "Data ref: Smith et al, 2001" or "Data ref: NCBI Sequence Read Archive PRJNA342805, 2017". In the Reference list, data citations must be labeled with "[DATASET]". A data reference must provide the database name, accession number/identifiers, and a resolvable link to the landing page from which the data can be accessed at the end of the reference. Further instructions are available at: <https://www.embopress.org/page/journal/14602075/authorguide#referencesformat>.
- Please include in section "Mice" of your Methods the reference number for approval of your experiments involving animals by the authority granting ethics approval.
- Appendix Tables should be renamed to "Dataset EV1-EV3" with their legends uploaded in a separate tab/sheet in each corresponding Excel file instead of being zipped together. Source file names, titles, legends, and manuscript callouts all need to be updated to "Dataset EV1-EV3".
- Please include the title of the manuscript following the heading "Appendix for" on the first page of your Appendix PDF file.
- The data in your Figure 1B represent mean percentage values from two biological replicates (n=2). As per our journal's policy, no statistics can be calculated and shown in the manuscript when n=2. Instead, please show the individual percentage values from each replicate.
- Please provide the exact p values in the legend of Figure 1C, if they are available.
- During our routine figure checks, we detected cell reuse (not detailed in the legends) between Figure 5A and Appendix Figure S4D. We kindly ask you to double-check these figures and correct them if necessary or, if the reuse is justified and intentional, mention it in the figure legends.
- The manuscript's section order should be corrected as follows: Title page - Abstract & Keywords - Introduction - Results - Discussion - Methods - Data Availability - Acknowledgements - Disclosure and Competing Interests Statement - References - Figure Legends - main Tables with legends (if there are any) - Expanded View Figure Legends.

Please also note that as part of the EMBO publications' Transparent Editorial Process, The EMBO Journal publishes online a Peer Review File along with each accepted manuscript. This File will be published in conjunction with your paper and will include the referee reports, your point-by-point response and all pertinent correspondence relating to the manuscript. You can opt out of this by letting the editorial office know (contact@embojournal.org). If you do opt out, the Peer Review File link will point to the following statement: "No Peer Review File is available with this article, as the authors have chosen not to make the review process public in this case."

We look forward to seeing a final version of your manuscript as soon as possible. Please let us know if you have any questions and use this link to submit your revision: <https://emboj.msubmit.net/cgi-bin/main.plex>.

Best regards,

Ioannis

Referee #2:

The manuscript has improved significantly. However, there are two clear weaknesses and a few instances of overstatement. I strongly suggest that the authors address these weaknesses in the discussion. If they prefer not to tone down the overstatements, they should provide experimental support.

1. The mechanism by which Prdm12 represses Dach1 remains unclear, which represents a significant gap in the study. While addressing this may not be necessary, acknowledging this weakness in the discussion would be informative.
2. The most important weakness is the emphasis on the broad-to-restricted expression pattern of Dach1, which may still be overstated given the similar phenotypes observed in WD1 cKO and AD1 cKO mice. The authors could perform additional experiments to support the biological importance of the broad-to-restricted expression pattern. Alternatively, I suggest they tone down this claim.

Overstatements:

Overstatement 1:

"Depletion of Dach1 during its broad or restricted expression period predominantly alters gene expression in developing TrkB+ mechanosensory neurons. This observation argues for an uncoupling between the initiation of Dach1 expression in early post-mitotic somatosensory neurons and its functional turn-on when restricted to post-mitotic TrkB+ neurons."

The observation does not "argue" for an "uncoupling." To be uncoupled, the authors would need to assume that Dach1 is non-functional beyond this phase. However, the study provides no data that directly test its function during early expression.

Overstatement 2:

"These results demonstrate the importance of compartmentalizing the expression of Dach1 in developing post-mitotic neurons and further reflect that its functional activation may result from a temporal developmental switch during the process of somatosensory neurogenesis."

The meaning of "developmental switch" is unclear. Is there a clear functional transition? If so, the authors provide no mechanistic evidence to support how or why Dach1 becomes active in TrkB+ neurons. This statement also implies that Dach1 was inactive before this stage, but the authors offer no experimental support for either scenario.

Overstatement 3:

"Altogether, these observations argue that the expression of Dach1 must become subtype-restricted as soon as this transcription factor finds itself in an environment amenable to triggering its functional activity."

This statement strongly suggests necessity, but the evidence primarily shows defects arising from misexpression. These defects could be due to overexpression levels rather than a requirement for subtype restriction.

Referee #3:

I acknowledge that the authors have effectively and comprehensively addressed all my previous concerns regarding their study on the transcription factor Dach1 and its role in somatosensory neuron development. The revisions have satisfactorily clarified the temporal refinement of Dach1 expression and its functional implications. Based on the thorough investigation and revisions presented, I endorse this paper for publication.

RESPONSE TO REFEREE

Referee #2:

The manuscript has improved significantly. However, there are two clear weaknesses and a few instances of overstatement. I strongly suggest that the authors address these weaknesses in the discussion. If they prefer not to tone down the overstatements, they should provide experimental support.

1. The mechanism by which Prdm12 represses Dach1 remains unclear, which represents a significant gap in the study. While addressing this may not be necessary, acknowledging this weakness in the discussion would be informative.

We added a sentence in the discussion (line 533) acknowledging that the molecular mechanism by which Prdm12 represses Dach1 expression in nociceptors still needs to be resolved.

2. The most important weakness is the emphasis on the broad-to-restricted expression pattern of Dach1, which may still be overstated given the similar phenotypes observed in WD1 cKO and AD1 cKO mice. The authors could perform additional experiments to support the biological importance of the broad-to-restricted expression pattern. Alternatively, I suggest they tone down this claim.

We changed the title of the discussion section “The broad-to-restricted expression of Dach1 is tightly linked to its functional turn on” at line 425 by “The broad-to-restricted expression of Dach1 associates with its function identified in TrkB⁺ LTMR.”

To the sentence “We report that depletion of *Dach1* during its broad or restricted expression period predominantly alters gene expression in developing TrkB⁺ mechanosensory neurons” at line 438 we now specify that this gene expression alteration was assessed in both cases by the end of embryogenesis. We also specified at line 441 that genes were found misregulated at embryonic stage E18.5.

We added a sentence at line 443 to acknowledge that early transient transcriptional defects that are not reflected in dorsal root ganglia by the end of embryogenesis may occur when depleting Dach1 during its broad expression period. We further specify in this sentence that the broad-to-restricted expression to which we refer in our manuscript specifically corresponds to the one defined by Sharma et al in their former study (PMID: 31915380).

Overstatements:

Overstatement 1:

"Depletion of Dach1 during its broad or restricted expression period predominantly alters gene expression in developing TrkB⁺ mechanosensory neurons. This observation argues for an uncoupling between the initiation of Dach1 expression in early post-mitotic somatosensory neurons and its functional turn-on when restricted to post-mitotic TrkB⁺ neurons."

The observation does not "argue" for an "uncoupling." To be uncoupled, the authors would need to assume that Dach1 is non-functional beyond this phase. However, the study provides no data that directly test its function during early expression.

We tuned down this statement and removed from the sentence the terms “argue” and “uncoupling”. We rephrased it to further highlight its hypothetical nature. The sentence (line 438) now reads: “This observation suggests a discrepancy between the initiation of Dach1 expression in early post-mitotic somatosensory neurons and its functional turn-on when restricted to post-mitotic TrkB⁺ neurons”.

Overstatement 2:

"These results demonstrate the importance of compartmentalizing the expression of Dach1 in developing post-mitotic neurons and further reflect that its functional activation may result from a temporal developmental switch during the process of somatosensory neurogenesis."

The meaning of "developmental switch" is unclear. Is there a clear functional transition? If so, the authors provide no mechanistic evidence to support how or why Dach1 becomes active in TrkB⁺ neurons. This statement also implies that Dach1 was inactive before this stage, but the authors offer no experimental support for either scenario.

We now tuned down this statement and remove the reference to an undescribed developmental switch. This sentence, found at line 488 now reads “These results may represent a requirement to compartmentalize the expression of Dach1 during the process of somatosensory neurogenesis.”

Overstatement 3:

"Altogether, these observations argue that the expression of Dach1 must become subtype-restricted as soon as this transcription factor finds itself in an environment amenable to triggering its functional activity."

This statement strongly suggests necessity, but the evidence primarily shows defects arising from misexpression. These defects could be due to overexpression levels rather than a requirement for subtype restriction.

The statement was tuned down to highlight its hypothetical nature. The sentence, found at line 490, now reads: “Dach1 may need to become subtype restricted when it finds itself in a environment where its expression would result in a functional activity detrimental for the developing sensory neuron”

RESPONSE TO EDITORIAL REQUESTS

- You can provide two more keywords (up to five can be listed in total) after the Abstract of your revised manuscript, if you wish.

We now added “Dorsal Root Ganglia” to the keywords provided.

- Thank you for depositing your RNA-sequencing and ChIP-sequencing datasets to appropriate databases and for providing the information in your Data availability statement. The reviewer tokens can now be removed from this statement. Please make sure that the datasets will be publicly available (from the provided URLs) at the time of publication.

The datasets have been made publicly available.

- Our journal encourages inclusion of data citations in the reference list to directly cite datasets that were obtained from public databases. Data citations in the article text are distinct from normal bibliographical citations and should directly link to the database records from which the data can be accessed. In the main text, data citations are formatted as follows: "Data ref: Smith et al, 2001" or "Data ref: NCBI Sequence Read Archive

PRJNA342805, 2017". In the Reference list, data citations must be labeled with "[DATASET]". A data reference must provide the database name, accession number/identifiers, and a resolvable link to the landing page from which the data can be accessed at the end of the reference. Further instructions are available at: <https://www.embopress.org/page/journal/14602075/authorguide#referencesformat>.

We have added two dataset references to the manuscript:

- Kupari J & Ernfors P (2023) Gene Expression Omnibus GSE139088 (<https://www.ncbi.nlm.nih.gov/geo/query/acc.cgi?acc=GSE139088>), Sequence Read Archive SRP135960 (<https://www.ncbi.nlm.nih.gov/sra/SRP135960>), <https://ernforsgroup.shinyapps.io/MouseDRGNeurons/> [DATASET]
- Sharma N, Flaherty K, Lezgiyeva K, Wagner DE, Klein AM & Ginty DD (2020) Gene Expression Omnibus GSE139088 (<https://www.ncbi.nlm.nih.gov/geo/query/acc.cgi?acc=GSE139088>), https://kleintools.hms.harvard.edu/tools/springViewer_1_6_dev.html?datasets/Sharma2019/all [DATASET]

- Please include in section "Mice" of your Methods the reference number for approval of your experiments involving animals by the authority granting ethics approval.

The reference number LA1500474 is now mentioned in the appropriate section.

- Appendix Tables should be renamed to "Dataset EV1-EV3" with their legends uploaded in a separate tab/sheet in each corresponding Excel file instead of being zipped together. Source file names, titles, legends, and manuscript callouts all need to be updated to "Dataset EV1-EV3".

The tables were renamed, the legend was added in the second sheet of each Excel file and the names of the tables were updated in the manuscript.

- Please include the title of the manuscript following the heading "Appendix for" on the first page of your Appendix PDF file.

The title of the manuscript was added to the first page of the Appendix PDF file.

- The data in your Figure 1B represent mean percentage values from two biological replicates (n=2). As per our journal's policy, no statistics can be calculated and shown in the manuscript when n=2. Instead, please show the individual percentage values from each replicate.

The plotted data already represent the individual percentage values from each replicate but the legend was misleading. The legend was thus corrected to remove the word "mean" from the sentence "... representing the mean percentage of Islet1⁺ neurons co-expressing..."

- Please provide the exact p values in the legend of Figure 1C, if they are available.

The p values in Figure 1C were calculated using Graphpad Prism. This software does not provide p values beyond four decimal places. We are therefore unfortunately not able to provide the exact values for this figure.

- During our routine figure checks, we detected cell reuse (not detailed in the legends) between Figure 5A and Appendix Figure S4D. We kindly ask you to double-check these

figures and correct them if necessary or, if the reuse is justified and intentional, mention it in the figure legends.

Despite we acknowledge that we did not initially notice the reuse of these panel figures, they do not interfere with the conclusions drawn in these figures as the pictures indeed came from an embryo that was both used to validate Dach1 overexpression in Advillin^{Cre/+}; Rosa26^{Dach1/+} E18.5 dorsal root ganglia (Appendix S4D) and included in the analysis to demonstrate that no change of Islet1+ neurons is detected between 18.5 control and Advillin^{Cre/+}; Rosa26^{Dach1/+} conditions (Figure 5A). We therefore mentioned in the legend of each figure that the picture was also used in the other figure as requested.

- The manuscript's section order should be corrected as follows: Title page - Abstract & Keywords - Introduction - Results - Discussion - Methods - Data Availability - Acknowledgements - Disclosure and Competing Interests Statement - References - Figure Legends - main Tables with legends (if there are any) - Expanded View Figure Legends.

We modified the manuscript to comply with the requested section order.

Dear Simon,

Congratulations on an excellent work! I am very pleased to inform you that your manuscript has now been accepted for publication in The EMBO Journal. Thank you for your thorough responses to the referees' comments, and for addressing our editorial and formatting requests.

If you have any questions, please do not hesitate to contact the Editorial Office. Thank you for your contribution to The EMBO Journal. Working with you has been a pleasure!

Best regards,

Ioannis
